# Diffusion-RainbowPA: Improvements Integrated Preference Alignment for Diffusion-based Text-to-Image Generation

**Haoyuan Sun**                                        *sun-hy23@mail.tsinghua.edu.cn*
*Tsinghua University*

**Bin Liang**                                          *bin.liang@uts.edu.au*
*University of Technology Sydney*

**Bo Xia**[*]                                          *xiab21@mails.tsinghua.edu.cn*
*Tsinghua University*

**Jiaqi Wu**                                           *wu-jq24@mails.tsinghua.edu.cn*
*Tsinghua University*

**Yifei Zhao**                                         *zhaoyife24@mails.tsinghua.edu.cn*
*Tsinghua University*

**Kai Qin**                                            *tank24@mails.tsinghua.edu.cn*
*Tsinghua University*

**Yongzhe Chang**                                      *changyongzhe@sz.tsinghua.edu.cn*
*Tsinghua University*

**Xueqian Wang**[*]                                    *wang.xq@sz.tsinghua.edu.cn*
*Tsinghua University*

**Reviewed on OpenReview:** `https://openreview.net/forum?id=KY0TSY2bx8`

## Abstract

Although rapidly increasing capabilities of text-to-image (T2I) models have profound implications across various industries, they concurrently suffer from numerous shortcomings, necessitating the implementation of effective alignment strategies with human preference. Diffusion-DPO and SPO have emerged as robust approaches for aligning diffusion-based T2I models with human preference feedback. However, they tend to suffer from text-image misalignment, aesthetic overfitting and low-quality generation. To tackle such matters, we improve the alignment paradigm through a tripartite perspective, which are the calibration enhancement (Calibration Enhanced Preference Alignment), the overfitting mitigation (Identical Preference Alignment, Jensen-Shannon Divergence Constraint) and the performance optimization (Margin Strengthened Preference Alignment, SFT-like Regularization). Furthermore, combining them with the step-aware preference alignment paradigm, we propose the Diffusion-RainbowPA, a suite of total six improvements that collectively improve the alignment performance of Diffusion-DPO. With comprehensive alignment performance evaluation and comparison, it is demonstrated that Diffusion-RainbowPA outperforms current state-of-the-art methods. We also conduct ablation studies on the introduced components that reveal incorporation of each has positively enhanced alignment performance.

---

[*]Corresponding authors: Bo Xia, Xueqian Wang

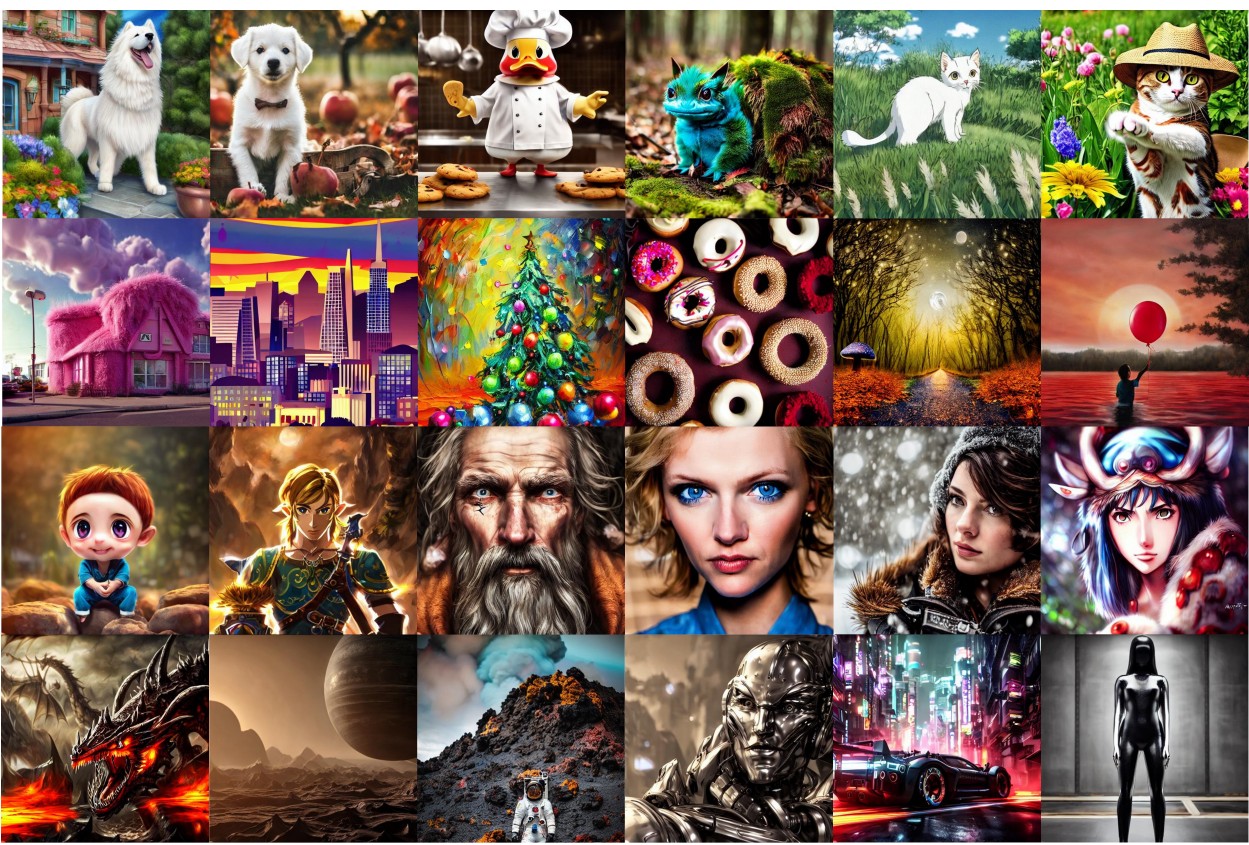

Figure 1: In this study, we propose the Diffusion-RainbowPA, a method that integrates six improvement components, collectively designed to enhance the alignment performance of Diffusion-DPO (Wallace et al., 2024). As shown in samples above, after the fine-tuning process on Stable Diffusion v1.5 model, our method has yielded images that possess an extraordinarily high level of visual allure and text alignment.

## 1 Introduction

*"The whole is greater than the sum of its parts."*

—— *Aristotle*

As generative capabilities of diffusion-based text-to-image (T2I) models advance, the challenge of aligning them with human preference has garnered extensive attention within the community. In relatively early works, alignment was commonly achieved through supervised fine-tuning (SFT) or Reinforcement Learning from Human Feedback (RLHF) (Black et al., 2024; Fan et al., 2024). While these methods are effective, they are hindered by convoluted implementation process. With notable success of Direct Preference Optimization (DPO) (Rafailov et al., 2024) in aligning Large Language Models (LLMs), preference-based alignment for T2I diffusion models offers a promising methodology that eliminates the need of reward modeling, thus streamlining training process. Diffusion-DPO (Wallace et al., 2024) and D3PO (Yang et al., 2024a) take the pioneering step for scaling of diffusion model alignment methods, introducing innovative alignment paradigm that learns from human preferences.

In the past year, numerous studies have been developed based on the paradigm (detailed in Section 2 and Appendix B). In particular, the recent SPO (Liang et al., 2024) introduces step-aware preference paradigm to address the misalignment between preference label and denoising performance at each stage. Nonetheless, there are still limitations persist. Firstly, as demonstrated in Table 1-4, execution of human preference alignment could worsen the misalignment between text and image. Furthermore, as depicted in Figure 2, they often tend to suffer from aesthetic overfitting and low-quality generation.

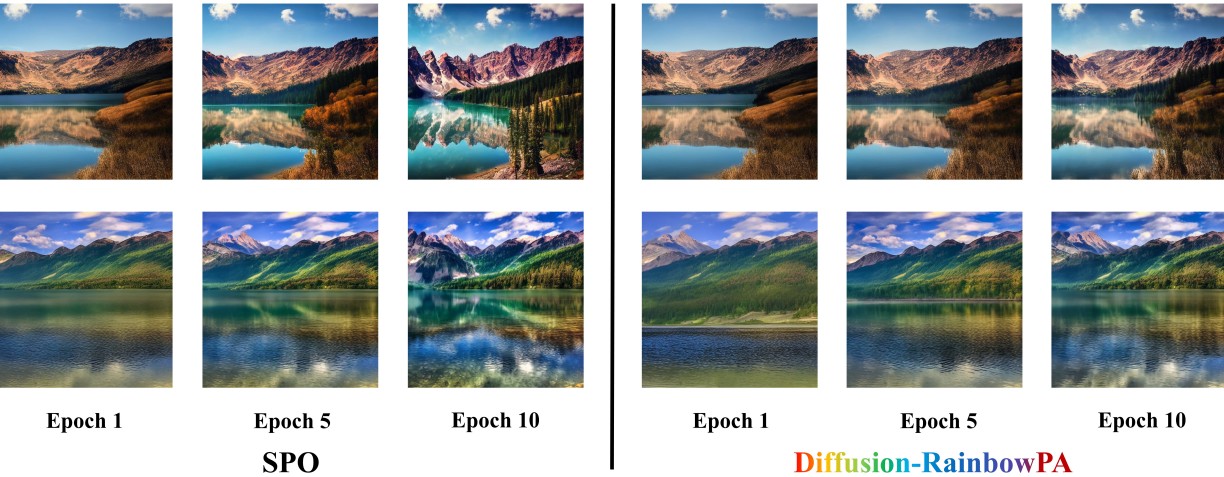

Figure 2: In the alignment training process over 10 epochs, generative examples of SPO (Liang et al., 2024) (*left*) and Diffusion-RainbowPA (*right*) with the prompt of "A beautiful lake". It can be observed that SPO leads to aesthetic overfitting, resulting in scenes with an unauthentic presence of snowy mountains adjacent to the lake, accompanied by chaotic coloration and excessive glare, thereby degrading quality of generated images. Conversely, Diffusion-RainbowPA has effectively mitigated such issues, providing an authentic depiction of scene and a natural rendering of color and light.

To address such issues, we improve alignment paradigm of diffusion-based T2I models from three aspects: the calibration enhancement (Section 4.2), the overfitting mitigation (Section 4.3) and the performance optimization (Section 4.4). In order to achieve such goals, we conduct an exhaustive investigation into the current alignment methods for LLMs (Wang et al., 2024; Winata et al., 2024) and carefully study their feasibility for adaptation to diffusion models (DMs). For the purpose of calibration enhancement, we introduce the Calibration Enhancement Preference Alignment (CEPA), introducing the calibration terms between the scaled ground-truth reward and the implicit reward function. In order to mitigate the issue of overfitting, we introduce the Identical Preference Alignment (IPA) and the Jensen-Shannon Divergence Constraint. Identical Preference Alignment (Section 4.3.1) skillfully circumvents the Bradley-Terry modeling assumption and employs an identical mapping to the preference function, thereby mitigating alignment preference overfitting. Jensen-Shannon Divergence Constraint (Section 4.3.2) substitutes Kullback-Leibler divergence with Jensen-Shannon divergence to stabilize alignment training and alleviate overfitting. Moreover, analysis of DPO-based methods for diffusion-based T2I models from the contrastive loss perspective shows that they only consider the "dissimilarity" part. Based on such observation, we introduce two performance optimization techniques: the Margin Strengthened Preference Alignment (MSPA) and the SFT-like Regularization. To accelerate increase of the diffusion win ratio and the margin between ratios, we introduce the Margin Strengthened Preference Alignment (Section 4.4.1). Furthermore, the SFT-like Regularization (Section 4.4.2) is proposed, which introduces a term to improve the impact of positive items. Based on the aforementioned five improving components that grounded in three perspectives, further integrating the step-aware preference alignment paradigm (Section 4.1), we propose the Diffusion-RainbowPA (Section 4.5), a novel method that integrates total six improvements on the Diffusion-DPO.

In the experiment evaluation, we utilize the Stable Diffusion v1.5 (Rombach et al., 2022) as our benchmark model, which is widely recognized and frequently utilized within the domain of T2I model alignment (encompassing both preference-based and RL-based methodologies). We evaluate the alignment performance with four published automatic metrics: {VQAScore (Lin et al., 2025), CLIPScore (Radford et al., 2021), HPS-V2 (Wu et al., 2023) and ImageReward (Xu et al., 2024)} on four zero-shot datasets: {GenEval (Ghosh et al., 2024), T2I-CompBench++ (Huang et al., 2025), GenAI-Bench (Li et al., 2024a), and DPG-Bench (Hu et al., 2024)}. Results (Section 5.2) indicate that alignment performance of Diffusion-RainbowPA outperforms current state-of-the-art (SOTA) methods. In Figure 1, we show generation examples of the aligned model that utilizing Diffusion-RainbowPA. Further ablation study (Section 5.3) on the introduced components

of Diffusion-RainbowPA demonstrates that each component has a positive impact on the improvement of alignment performance.

Our contributions are summarized as follows:

1. We point out that current existing SOTA methods still tend to suffer from text-image misalignment, aesthetic overfitting and low-quality generation.

2. To address such issues, we improve the alignment paradigm through a tripartite perspective: the calibration enhancement, the overfitting mitigation and the performance optimization.

3. We propose the Diffusion-RainbowPA, a novel method combining total six improvements on Diffusion-DPO, which outperforms current SOTA methods.

## 2  Related Work

Fueled by resounding success of Direct Preference Optimization (DPO) (Rafailov et al., 2024), which has revolutionized alignment processes for LLMs by obviating the requirement for explicit reward model, Diffusion-DPO (Wallace et al., 2024) and D3PO (Yang et al., 2024a) have taken the forefront in aligning diffusion-based T2I models with human preferences. Amid their remarkable success, several subsequent works (Yang et al., 2024b; Li et al., 2024b; Gu et al., 2024; Hong et al., 2024) have expanded upon their foundation. For instance, SPIN-Diffusion (Yuan et al., 2024), empowers the model to outperform its predecessors, thereby fostering an iterative cycle of continuous self-improvement. In the work (Liang et al., 2024), it is emphasized that there is a discrepancy between the preference for the final image and the performance at each denoising step. To tackle such challenge, a step-aware preference strategy is effectively deployed. Recently, in the study (Sun et al., 2025b), the authors suggest that substituting Jensen-Shannon divergence for Kullback-Leibler divergence can significantly improve alignment performance. SePPO (Zhang et al., 2024) employs the concept of online sampling, leveraging previously saved checkpoints as reference models, and has become one of the state-of-the-art methods. We provide more details about diffusion-based T2I models in Appendix A and offer a more in-depth and detailed discussion on diffusion-based T2I alignment in Appendix B. Although they have achieved promising results, challenges remain, including text-image misalignment, aesthetic overfitting and low-quality generation. This study adopts an integrated approach to address them concurrently.

## 3  Preliminaries

In this part, we provide a brief description on the Diffusion-DPO and the SPO for readers who are unfamiliar with diffusion-based text-to-image alignment.

### 3.1  Diffusion-DPO

Within framework of diffusion models, Diffusion-DPO (Wallace et al., 2024) extends the optimization objective from $p_\theta(x_0|c)$ to $p_\theta(x_{0:T}|c)$. By incorporating latent variables $x_{1:T}$, an implicit reward model is established across the entire sequence. Based on this, the optimization objective pertaining to the conditional distribution can be formulated as follows:

$$\mathcal{L}(\theta) = -\mathbb{E}_{\substack{(c,x_0^w,x_0^l)\sim\mathcal{D},\\ x_{1:T}^w\sim p_\theta(x_{1:T}^w|x_0^w,c),\\ x_{1:T}^l\sim p_\theta(x_{1:T}^l|x_0^l,c)}} \log\sigma\left(\beta\log\frac{p_\theta(x_{0:T}^w|c)}{p_{\text{ref}}(x_{0:T}^w|c)} - \beta\log\frac{p_\theta(x_{0:T}^l|c)}{p_{\text{ref}}(x_{0:T}^l|c)}\right), \quad (1)$$

where a fixed dataset $\mathcal{D} = \{(c,x_0^w,x_0^l)\}$ is employed with $c$ denoting the input prompt, $x_0^w$ indicating the human preferred generation and $x_0^l$ signifying the dispreferred one; $p_\theta(x_{0:T}|c)$ represents the reverse process parametrization, $p_{\text{ref}}(x_{0:T}|c)$ pertains to that of the reference model; $\beta$ is the measure of regularization intensity and $\sigma(\cdot)$ is the Sigmoid function.

Through application of reverse decompositions to $p_\theta$ and $p_{\text{ref}}$, leveraging Jensen's inequality and convexity of function $-\log \sigma$, the authors derive the following upper bound on optimization:

$$\mathcal{L}(\theta) \leq -\mathbb{E}_{\substack{(c,x_0^w,x_0^l)\sim D, t\sim\mathcal{U}(0,T), \\ x_{t-1,t}^w \sim p_\theta(x_{t-1,t}^w|c,x_0^w), \\ x_{t-1,t}^l \sim p_\theta(x_{t-1,t}^l|c,x_0^l)}} \log \sigma \left( \beta T \log \frac{p_\theta(x_{t-1}^w|c,x_t^w)}{p_{\text{ref}}(x_{t-1}^w|c,x_t^w)} - \beta T \log \frac{p_\theta(x_{t-1}^l|c,x_t^l)}{p_{\text{ref}}(x_{t-1}^l|c,x_t^l)} \right). \tag{2}$$

## 3.2 Step-aware Preference Optimization

In the work (Liang et al., 2024), it is noted that previous methods, such as Diffusion-DPO (Wallace et al., 2024) and D3PO (Yang et al., 2024a), solely evaluate the whole generation trajectory based on the final image $x_0$, assigning all intermediate states the same preference as $x_0$. Hence, based on such observation, the authors introduce Step-aware Preference Optimization (SPO) methodology. It primarily hinges on two key aspects. Firstly, during each denoising step, an image pair is determined from the generated image subset, and the winner and loser are thereby ascertained. To achieve the target, a step-aware preference model is trained to synchronize with denoising performance. Furthermore, a step-wise resampler is designed to eliminate trajectory-level dependencies. Merging objectives across all $T$ timesteps, the ultimate objective function for SPO training can be derived as follows:

$$\mathcal{L}(\theta) = -\mathbb{E}_{\substack{t\sim\mathcal{U}[1,T],c\sim p(c),x_T\sim\mathcal{N}(0,I), \\ (x_{t-1}^w,x_{t-1}^l)\sim p_\theta(x_{t-1}|c,t,x_t)}} \log \sigma \left( \beta \log \frac{p_\theta(x_{t-1}^w|c,t,x_t)}{p_{\text{ref}}(x_{t-1}^w|c,t,x_t)} - \beta \log \frac{p_\theta(x_{t-1}^l|c,t,x_t)}{p_{\text{ref}}(x_{t-1}^l|c,t,x_t)} \right), \tag{3}$$

where the preference pair $(x_{t-1}^w, x_{t-1}^l)$, comprising the most preferred item $x_{t-1}^w$ and the most dispreferred item $x_{t-1}^l$ from the sampling set $\{x_{t-1}^1, x_{t-1}^2, ..., x_{t-1}^k\}$.

# 4 Method

Diffusion-DPO (Wallace et al., 2024) has emerged as a significant milestone in bridging the gap of alignment paradigm between large language models (LLMs) and diffusion models (DMs). Recently, numerous improvements have been proposed to enhance the preference-based alignment paradigm of DMs. However, they still tend to suffer from the issues of text-image misalignment, aesthetic overfitting and low-quality generation. Subsequently, to mitigate such issues, we improve the alignment paradigm with five additional components through a tripartite perspective: calibration enhancement (Section 4.2), overfitting mitigation (Section 4.3) and performance optimization (Section 4.4). Further integrating the step-aware preference alignment paradigm (Section 4.1), we propose the Diffusion-RainbowPA (Section 4.5), a suite of total six improvements integrated that collectively enhance alignment performance of Diffusion-DPO.

## 4.1 Step-aware Preference Alignment Paradigm

In Section 3.2, we conduct a brief review of SPO, which advocates the employment of a step-aware preference model and a step-wise resampler to facilitate precise step-aware supervision, while maintaining the same initialization throughout the subsequent denoising phase. Throughout our study, we consistently integrate the paradigm of step-aware preference and utilize the open-source step-aware preference model in our alignment framework.

Moreover, unlike original SPO, we abandon multi-sample setup utilized in step-wise resampler (which samples 4 images in SPO) and adopt a setup that samples 2 images. Such a choice is informed by the following observations, as depicted in the left side of Figure 2, it is found that SPO leads to overfitting in aesthetic rating. This is further corroborated by Table 6 of SPO (Liang et al., 2024), where, despite optimal aesthetic rating at the sampling number of 4, human preference ratings (HPS-V2 and ImageReward) are the lowest.

## 4.2 Calibration Enhancement

As previously mentioned, the issue of text-image misalignment still persists. Hence, we introduce the **C**alibration **E**nhancement **P**reference **A**lignment (CEPA) to address it. Firstly, we draw inspiration from

Cal-DPO (Xiao et al., 2024) and define the calibrated objective for diffusion-based preference optimization in Definition 1.

**Definition 1.** *If we have* $\beta \cdot \log \frac{p_\theta(x_{t-1}|c,t,x_t)}{p_{ref}(x_{t-1}|c,t,x_t)} = r(c, x_0)$, *we call that the estimated implicit reward* $\beta \cdot \log \frac{p_\theta(x_{t-1}|c,t,x_t)}{p_{ref}(x_{t-1}|c,t,x_t)}$ *for the sampling probability of DM* $p_\theta$ *is calibrated with the ground truth reward.*

Hence, we introduce the CEPA term with the purpose of constraining distance between the learned implicit reward and the ground-truth reward. Furthermore, it is observed that formula form of the term is similar to SPPO (Wu et al., 2024), which treats alignment problem as a constant-sum two-player game and approximate Nash equilibrium through iterative policy updates. Such observation further illustrates its effectiveness and inspires us to set the reward for preference feedback as: $r(c, x_0^w) = 1/2$ and $r(c, x_0^l) = -1/2$. In practice, the setting works well. Then, we show terms of the CEPA as follows:

$$\mathcal{L}_{\text{CEPA}}(\theta) = \mathbb{E}_{\substack{t \sim \mathcal{U}[1,T], c \sim p(c), x_T \sim \mathcal{N}(0,I), \\ (x_{t-1}^w, x_{t-1}^l) \sim p_\theta(x_{t-1}|c,t,x_t)}} \left[ \beta \log \frac{p_\theta(x_{t-1}^w|c,t,x_t^w)}{p_{\text{ref}}(x_{t-1}^w|c,t,x_t^w)} - \frac{1}{2} \right]^2 + \left[ \beta \log \frac{p_\theta(x_{t-1}^l|c,t,x_t^l)}{p_{\text{ref}}(x_{t-1}^l|c,t,x_t^l)} + \frac{1}{2} \right]^2. \quad (4)$$

From Equation (4), we can see that CEPA foucs on increasing the discrepancy between the diffusion win ratio and the diffusion lose ratio to a scale of 1. Furthermore, it endeavors to adjust the diffusion win ratio close to $1/2$ and the diffusion lose ratio close to $-1/2$ for simultaneously increasing log-likelihood of the preferred item and decreasing that of the dispreferred one.

## 4.3 Overfitting Mitigation

Large diffusion models that have undergone extensive data pre-training are prone to overfitting during and alignment process (Gao et al., 2023; Clark et al., 2024; Kim et al., 2025). In this study, we effectively mitigate propensity for overfitting during the alignment process by employing the **I**dentical **P**reference **A**lignment (IPA) and the Jensen-Shannon Divergence Constraint.

### 4.3.1 Identical Preference Alignment

In the work (Azar et al., 2024), it is proposed that overfitting issue is partly due to the replacement of pairwise preferences with a pointwise reward model, as implemented in Bradley-Terry (Bradley & Terry, 1952) model, encountering challenges when preferences are predictable or nearly predictable. Hence, in instances where the preference leans towards determinism, as is often the case in image generation, discrepancy between reward functions would asymptotically tend towards infinity, thereby substantially diminishing the effectiveness of divergence constraint enforced by $\beta$. Based on such observation (Azar et al., 2024) propose the IPO to tackle such challenge, which adeptly sidesteps the Bradley-Terry modeling assumption concerning preferences and applies *an identical mapping to the preference function*. Furthermore, the very recent work (Sun et al., 2025c) suggests that such paradigm can mitigate overfitting in the alignment process of text-to-image models, leading to improved alignment performance and generation diversity. Hence, inspired by the idea of identical mapping, we introduce the **I**dentical **P**reference **A**lignment (IPA) that further extends it to diffusion-based T2I alignment and concurrently merges the step-aware preference alignment paradigm:

$$\mathcal{L}_{\text{IPA}}(\theta) = \mathbb{E}_{\substack{t \sim \mathcal{U}[1,T], c \sim p(c), x_T \sim \mathcal{N}(0,I), \\ (x_{t-1}^w, x_{t-1}^l) \sim p_\theta(x_{t-1}|c,t,x_t)}} \left[ \log \left( \frac{p_\theta(x_{t-1}^w|c,t,x_t^w)p_{\text{ref}}(x_{t-1}^l|c,t,x_t^l)}{p_\theta(x_{t-1}^l|c,t,x_t^l)p_{\text{ref}}(x_{t-1}^w|c,t,x_t^w)} \right) - \frac{1}{2\beta} \right]^2, \quad (5)$$

where $\beta$ is the regularization intensity.

### 4.3.2 Jensen-Shannon Divergence Constraint

Kullback-Leibler (KL) divergence has been typically chosen as the divergence constraint for policy training in traditional methods such as Diffusion-DPO and SPO. In the work (Sun et al., 2025b), it extends the KL divergence constraint to $f$-divergence, including the Forward KL divergence, the Reverse KL divergence, the $\alpha$-divergence and the Jensen-Shannon (JS) divergence. Their detailed analysis and comprehensive experiments

show that JS divergence exhibits superior generative diversity as well as better alignment performance compared to KL divergence. Generalized formula under the constraint of $f$-divergence is presented as:

$$\mathcal{L}_f(\theta) = -\mathbb{E}_{\substack{t\sim\mathcal{U}[1,T],c\sim p(c),x_T\sim\mathcal{N}(0,I),\\(x_{t-1}^w,x_{t-1}^l)\sim p_\theta(x_{t-1}|c,t,x_t)}} \log\sigma\left(\beta f'\left(\frac{p_\theta(x_{t-1}^w|c,t,x_t)}{p_{\text{ref}}(x_{t-1}^w|c,t,x_t)}\right) - \beta f'\left(\frac{p_\theta(x_{t-1}^l|c,t,x_t)}{p_{\text{ref}}(x_{t-1}^l|c,t,x_t)}\right)\right), \quad (6)$$

where $f'(\cdot)$ represents the derivatives of generator function $f(\cdot)$. Upon further comparing Equation (6) and Equation (3), it can be observed that generalization from KL divergence to $f$-divergence only requires the substitution from $\log x$ to $f'(x)$, where $x$ represents the win ratio and the loss ratio. Therefore, if this component is activated, for JS divergence, $f(x) = x\log\frac{2x}{x+1} + \log\frac{2}{x+1}$ and $f'(x) = \log\frac{2x}{x+1}$, following substitution is needed:

$$\log\frac{p_\theta(x_{t-1}^*|c,x_t^*)}{p_{\text{ref}}(x_{t-1}^*|c,x_t^*)} \longrightarrow \log\left[2\cdot\frac{p_\theta(x_{t-1}^*|c,x_t^*)}{p_{\text{ref}}(x_{t-1}^*|c,x_t^*)}\Big/\left(1+\frac{p_\theta(x_{t-1}^*|c,x_t^*)}{p_{\text{ref}}(x_{t-1}^*|c,x_t^*)}\right)\right] \quad (7)$$

## 4.4 Performance Optimization

In practice, aligned diffusion models still suffer from low-quality generation. In this section, we aim to achieve further performance optimization.

Let's begin by re-examining DPO-based methods from the perspective of *contrastive loss functions* (Sun et al., 2025d; Lv et al., 2025). With the purpose of minimizing distances of similar items and maximizing distances of dissimilar ones, contrastive loss (Hadsell et al., 2006) has been widely applied in large model training, such as the CLIP model (Radford et al., 2021):

$$\mathcal{L}_{\text{con}} = -\Big[\underbrace{\sum\mathcal{D}(y_i,y_j)}_{\text{dissimilar items' distance}} - \lambda\cdot\underbrace{\sum\min\left(\mathcal{L}(y_i)-\epsilon,0\right)}_{\text{similar items' term}}\Big], \quad (8)$$

where $\mathcal{D}(y_i,y_j)$ is the distance between a positive item $y_i$ and a negative item $y_j$; $\lambda$ represents regularization intensity of positive items and $\epsilon$ is the introduced threshold. Define the dissimilar instances' distance between the positive (preferred) item $x_{t-1}^w$ and the negative (dispreferred) item $x_{t-1}^l$ as:

$$\mathcal{D} = \log\frac{p_\theta(x_{t-1}^w|c,t,x_t)}{p_{\text{ref}}(x_{t-1}^w|c,t,x_t)} - \log\frac{p_\theta(x_{t-1}^l|c,t,x_t)}{p_{\text{ref}}(x_{t-1}^l|c,t,x_t)} \quad (9)$$

Hence, it can be observed that conventional DPO framework incorporates only the dissimilarity component (distance between positive and negative items) within the framework of contrastive loss, while neglecting to consider the constituent element that are relevant to the similar items' term. In this part, we break the balance between positive and negative items in two ways to improve the alignment process's learning of positive items, i.e. the **M**argin **S**trengthened **P**reference **A**lignment (MSPA) and the SFT-like Regularization.

### 4.4.1 Margin Strengthened Preference Alignment

As shown in the Equation (3), we aim to maximize the diffusion win ratio while concurrently minimizing the diffusion lose ratio during the training phase. However, it is usually the case that both ratios decline concurrently to achieve margin expansion, with the diffusion lose ratio exhibiting a more pronounced rate of decrease. *Continuous decline of the diffusion win ratio is highly detrimental to the stability of training process and the safety of fine-tuned model.* Similar to LLM contexts (Yan et al., 2025), it will gradually fail to control alignment direction of the preferred items, ultimately devolving into simply unlearning the dispreferred items. Furthermore, such training process can lead to a dispersion effect on unseen responses, e.g. model's capacity of generating unseen content in images has risen dramatically, posing a substantial risk to generative safety.

Based on such observations and reflections, we would like to optimize the alignment process by targeting the following two objectives: increasing margin between the diffusion win ratio and the diffusion lose ratio; decelerating decrease and subsequently accelerating increase in the diffusion win ratio. In order to achieve

them simultaneously, a diffusion win ratio strengthened term is introduced within the $\sigma(\cdot)$ function of Equation (3), thus transforming it into the following form:

$$\mathcal{L}_{\text{MSPA}}(\theta) = -\mathbb{E}_{\substack{t\sim\mathcal{U}[1,T],c\sim p(c),x_T\sim\mathcal{N}(0,I),\\(x_{t-1}^w,x_{t-1}^l)\sim p_\theta(x_{t-1}|c,t,x_t)}}\log\sigma\left(\eta\beta\log\frac{p_\theta(x_{t-1}^w|c,t,x_t)}{p_{\text{ref}}(x_{t-1}^w|c,t,x_t)} + \beta\log\frac{p_\theta(x_{t-1}^w|c,t,x_t)}{p_{\text{ref}}(x_{t-1}^w|c,t,x_t)}\right.$$
$$\left. -\beta\log\frac{p_\theta(x_{t-1}^l|c,t,x_t)}{p_{\text{ref}}(x_{t-1}^l|c,t,x_t)}\right), \tag{10}$$

where $\eta$ is strengthening intensity of the margin and the diffusion win ratio.

Moreover, it is evident that the formulation presented in Equation (10) substantively disrupts equilibrium between the win ratio and the loss ratio, thus enabling the formula exhibits similar items' terms within the modified expression.

### 4.4.2 SFT-like Regularization

Based on the aforementioned observations and the gains achieved by modifications within MSPA, we further consider another common performance optimization method in LLM alignment: the SFT regularization (Yan et al., 2025). However, it's also worth noting that diffusion models actually exhibit a different paradigm compared to LLMs. Totally, there are three enhanced strategies: enhancing probability of preferred ($p_\theta$); enhancing log probability of preferred ($\log p_\theta$); enhancing log probability of preferred ratio ($\log\frac{p_\theta}{p_{\text{ref}}}$). According to the very recent work (Sun et al., 2025d), it is suggested that enhancing log probability of preferred ratio ($\log\frac{p_\theta}{p_{\text{ref}}}$) not only yields excellent effectiveness but also ensures robust training stability. Furthermore, we introduce a threshold and the minimum operation to align its form with that of similar items' term in Equation (8). While, note that this is not the SFT loss, we designate it as the SFT-like Regularization:

$$\mathcal{L}_{\text{SFT-L}}(\theta) = -\mathbb{E}_{\substack{t\sim\mathcal{U}[1,T],c\sim p(c),x_T\sim\mathcal{N}(0,I),\\(x_{t-1}^w,x_{t-1}^l)\sim p_\theta(x_{t-1}|c,t,x_t)}}\lambda\cdot\min\left(\log\frac{p_\theta(x_{t-1}^w|c,t,x_t)}{p_{\text{ref}}(x_{t-1}^w|c,t,x_t)},\epsilon\right), \tag{11}$$

where $\lambda$ is the enhancement intensity and $\epsilon$ is the threshold.

### 4.5 Diffusion-RainbowPA

Inspired by Aristotle's assertion that "the whole is greater than the sum of its parts", and further motivated by Rainbow (Hessel et al., 2018) and RainbowPO (Zhao et al., 2025), we propose combining these improvements. Building on the aforementioned improvements to Diffusion-DPO, encompassing five components through a tripartite perspective and further integrating the step-aware preference alignment paradigm, we introduce the Diffusion-RainbowPA: a comprehensive combination of six improvements to the Diffusion-DPO. Combining the formula in Equation (4), Equation (5), Equation (10) and Equation (11), we can obtain following form:

$$\mathcal{L}_{\text{total}}(\theta) = \mathcal{L}_{\text{IPA}}(\theta) + \mathcal{L}_{\text{SPPA}}(\theta) + \mathcal{L}_{\text{PEPA}}(\theta) + \mathcal{L}_{\text{MSPA}}(\theta) \tag{12}$$

Furthermore, with activation of the component in Section 4.3.2, specifically by executing transformation defined in Equation (7), we convert $\mathcal{L}_{\text{total}}(\theta)$ into $\mathcal{L}_{\text{Diffusion-RainbowPA}}(\theta)$, which serves as the alignment objective of Diffusion-RainbowPA:

$$\mathcal{L}_{\text{total}}(\theta) \xrightarrow[\text{in Equation (7)}]{\text{Transformation}} \mathcal{L}_{\text{Diffusion-RainbowPA}}(\theta) \tag{13}$$

Table 1: Comparison on the GenEval.

| GenEval | VQAS ↑ | CLIPS ↑ | HPS ↑ | IR ↑ |
|---|---|---|---|---|
| SD-1.5 | 61.85 | 33.77 | 26.93 | -0.200 |
| Diffusion-DPO | 63.68 | 34.52 | 27.17 | -0.030 |
| SPO | 61.22 | 33.41 | 27.30 | 0.083 |
| SPO (LoRA) | 61.38 | 33.40 | 27.30 | 0.081 |
| SPIN-Diffusion | 60.81 | 32.79 | 27.31 | -0.073 |
| SePPO | 63.10 | 33.94 | 27.35 | 0.003 |
| **Ours** | **68.41** | **35.01** | **27.45** | **0.202** |

Table 2: Comparison on the T2I-CompBench++.

| T2I-Comp++ | VQAS ↑ | CLIPS ↑ | HPS ↑ | IR ↑ |
|---|---|---|---|---|
| SD-1.5 | 60.71 | 32.06 | 26.55 | -0.245 |
| Diffusion-DPO | 62.61 | 32.66 | 26.78 | -0.089 |
| SPO | 62.73 | 31.19 | 26.87 | -0.023 |
| SPO (LoRA) | 62.77 | 31.19 | 26.87 | 0.021 |
| SPIN-Diffusion | 60.84 | 31.15 | 27.00 | -0.086 |
| SePPO | 63.10 | 32.30 | 27.03 | -0.006 |
| **Ours** | **67.60** | **33.48** | **27.15** | **0.230** |

Table 3: Comparison on the GenAI-Bench.

| GenAI-Bench | VQAS ↑ | CLIPS ↑ | HPS ↑ | IR ↑ |
|---|---|---|---|---|
| SD-1.5 | 59.50 | 33.62 | 26.94 | 0.237 |
| Diffusion-DPO | 60.15 | 34.11 | 27.15 | 0.376 |
| SPO | 59.52 | 32.40 | 27.27 | 0.360 |
| SPO (LoRA) | 59.48 | 32.41 | 27.27 | 0.359 |
| SPIN-Diffusion | 58.06 | 32.92 | 27.35 | 0.437 |
| SePPO | 61.11 | 34.09 | 27.48 | 0.536 |
| **Ours** | **62.58** | **34.47** | **27.53** | **0.593** |

Table 4: Comparison on the DPG-Bench.

| DPG-Bench | VQAS ↑ | CLIPS ↑ | HPS ↑ | IR ↑ |
|---|---|---|---|---|
| SD-1.5 | 76.28 | 33.42 | 26.31 | -0.187 |
| Diffusion-DPO | 77.13 | 34.01 | 26.51 | -0.036 |
| SPO | 75.61 | 32.13 | 26.73 | -0.087 |
| SPO (LoRA) | 75.60 | 32.12 | 26.73 | -0.091 |
| SPIN-Diffusion | 76.59 | 32.86 | 26.84 | 0.109 |
| SePPO | 78.30 | 34.09 | 26.85 | **0.175** |
| **Ours** | **79.18** | **34.72** | **26.95** | 0.166 |

## 5 Experiments

### 5.1 Experimental Settings

**Model and Dataset.** In the study, we utilize the Stable Diffusion v1-5 (SD-1.5) as our benchmark model, as it is the most widely applied model in both the RL-based alignment research group and the preference-based alignment research group. In selecting the training dataset, to ensure a fair comparison, we adopt the same dataset utilized by SPO (Liang et al., 2024), which consists of 4K randomly chosen prompts from the Pick-a-Pic V1 dataset.

**Hyperparameters.** We simultaneously set all terms in Equation (12) to share $\beta = 10$, corresponding to the SPO condition. Based on the tuning results reported in (Sun et al., 2025d), the positive enhancement intensity $\lambda$ in Equation (11) is set as 100 and the threshold as $\log 0.9$; for the MSPA term, the margin strengthening intensity $\eta$ is empirically set to 0.5.

**Evaluations.** To comprehensively verify the improvement in alignment performance, we adopt a total of four zero-shot alignment performance metrics, which are: the VQAScore (Lin et al., 2025) (abbreviated as **VQAS**), the CLIPScore (Radford et al., 2021) (abbreviated as **CLIPS**), the HPS-V2 (Wu et al., 2023) (abbreviated as **HPS**), and ImageReward (Xu et al., 2024) (abbreviated as **IR**). We provide more details on the evaluation metrics in Appendix C.1. In our selection of evaluation benchmark, we opt for a total of four widely utilized benchmark datasets in the area of text-to-image generalization: the GenEval (Ghosh et al., 2024), the T2I-CompBench++ (Huang et al., 2025), the GenAI-Bench (Li et al., 2024a), and the DPG-Bench (Hu et al., 2024). Unlike some previous work, our evaluation uses an unified approach: comparing the quality of generated images with the four zero-shot metrics described above, as opposed to contrasting them with their individual benchmarking results. We provide more details on the evaluation datasets in Appendix C.2.

### 5.2 Comparison with State-of-the-art Methods

In order to demonstrate the superiority of Diffusion-RainbowPA, we compare quantitative results with those of state-of-the-art methods, including RL-based methods (the SPIN-Diffusion (Yuan et al., 2024) and the

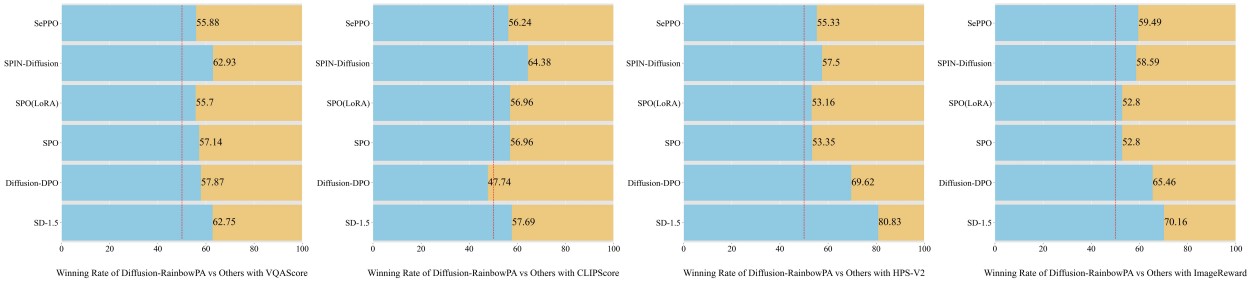

Figure 3: Winning rate from model feedback on the GenEval.

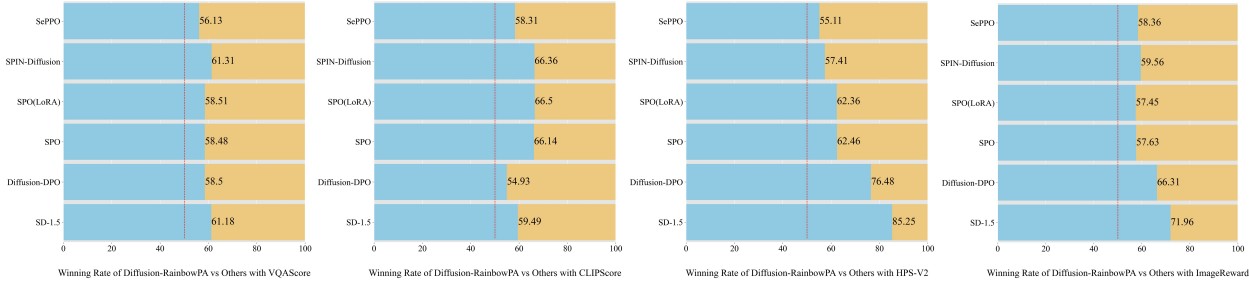

Figure 4: Winning rate from model feedback on the T2I-CompBench++.

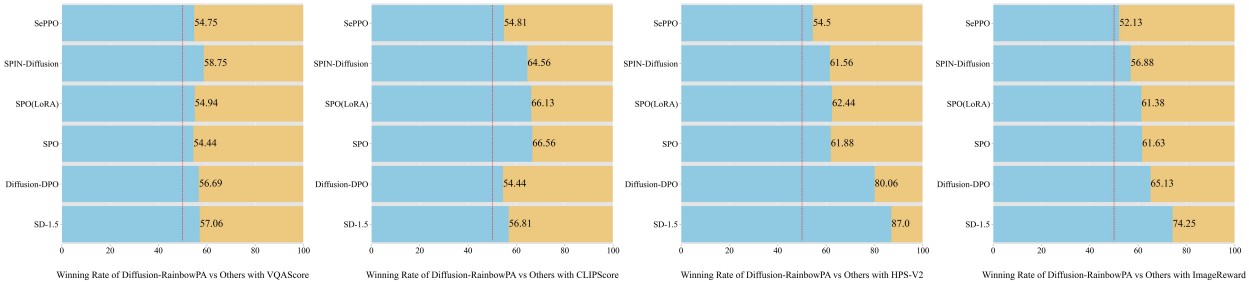

Figure 5: Winning rate from model feedback on the GenAI-Bench.

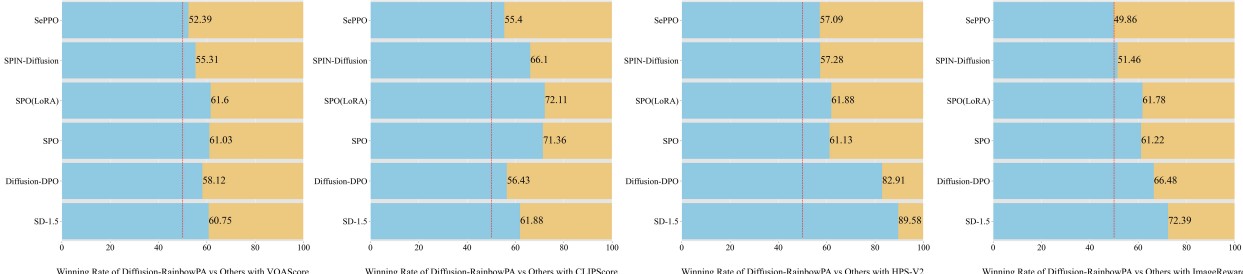

Figure 6: Winning rate from model feedback on the DPG-Bench.

SePPO (Zhang et al., 2024)) and preference-based methods (the Diffusion-DPO (Wallace et al., 2024) and the SPO (Liang et al., 2024)). We provide more details on the comparison methods in Appendix C.3. In Table 1, Table 2, Table 3, Table 4, we report quantitative comparison results across four zero-shot alignment metrics on the GenEval, the T2I-CompBench++, the GenAI-Bench, and the DPG-Bench, respectively. It is particularly noteworthy that improvement of Diffusion-RainbowPA in both text-image alignment and human preference alignment is unparalleled. Specifically, in the context wherein short prompts are employed (GenEval and T2I-CompBench++): on GenEval, our method achieves a VQAScore improvement of 6.56, while the best other SOTAs only achieves a maximum of 1.83; on T2I-CompBench++, our method improves the VQAScore by 6.89, while the highest among other SOTAs is only 2.39. Simultaneously, the results

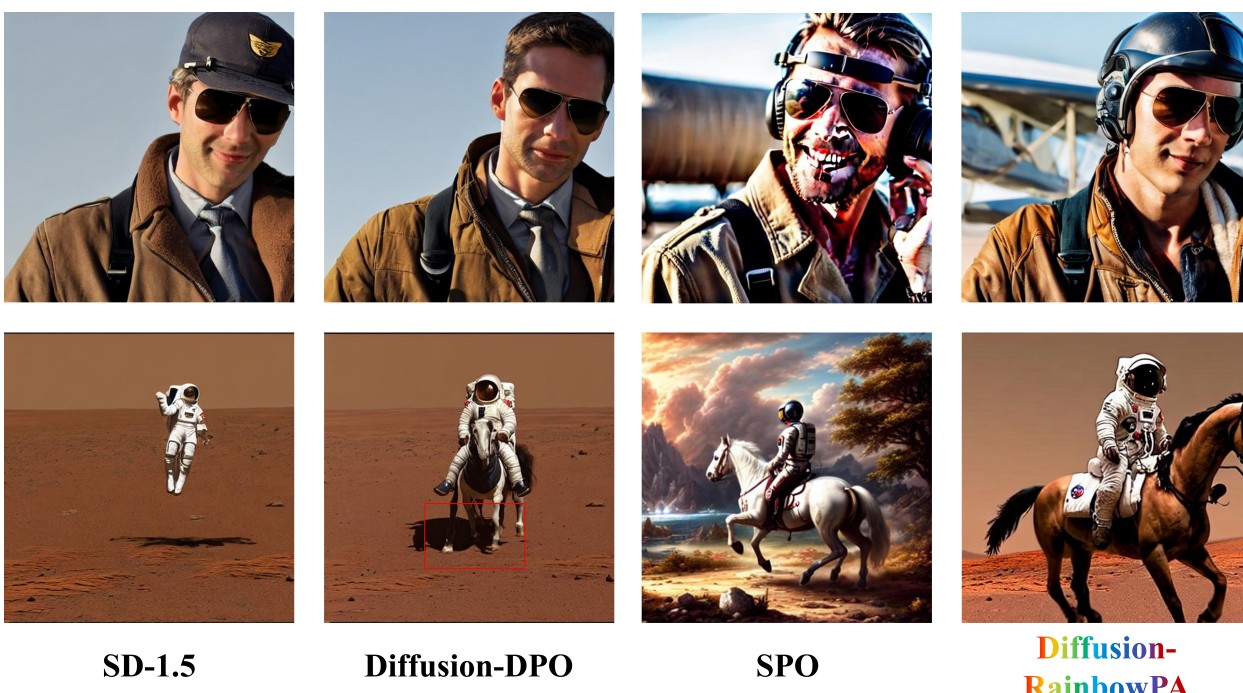

**SD-1.5**  **Diffusion-DPO**  **SPO**  **Diffusion-RainbowPA**

Figure 7: Qualitative comparison of Diffusion-RainbowPA with Diffusion-DPO and SPO. Top: "A **pilot** with aviator sunglasses."; Bottom: "A **photo** of an astronaut riding a **horse** on **Mars**." It can be observed that Diffusion-RainbowPA have achieved **superior text-image alignment** performance (aspects such as *scene*, *attribute*, and *style*), **alleviative overfitting** and **generation with higher quality**.

pertaining to long prompts are exciting: on GenAI-Bench, our method yields an improvement in VQAScore of 3.08, whereas the most superior SOTA attains a peak improvement of merely 1.61; on DPG-Bench, our method demonstrates an improvement in VQAScore of 2.9, but top-performing SOTAs merely achieve a maximum of 2.02. We also provide detailed quantitative results on the eight separate categories of T2I-CompBench++ in Appendix D. Furthermore, in Figure 3, Figure 4, Figure 5, Figure 6, we present the winning rate of Diffusion-RainbowPA against other algorithms based on model feedbacks (VQAScore, CLIPScore, HPS-V2 and ImageReward) on the GenEval, the T2I-CompBench++, the GenAI-Bench and the DPG-Bench, respectively. The results indicate that Diffusion-RainbowPA consistently achieves superior performance.

Furthermore, we display qualitative comparisons between Diffusion-RainbowPA and the SOTAs, as detailed in Figure 7. It can be observed that, Diffusion-RainbowPA exhibits superior text-image alignment performance, more realistic scene depiction, more precise attributes, and more expected generation style. For example, in the first line, our method generates a more accurate, aesthetically pleasing, and logically coherent output for the subject "pilot"; in the second line, our method is more accurate for scene ("Mars"), attribute (like leg number of "horse") and style ("photo").

## 5.3 Ablation Study

To further validate the improvement of alignment performance by each component, we conduct ablation study on the five introduced components with the aforementioned four benchmark datasets. In Table 5, Table 6, Table 7, Table 8, we show the alignment performance on the GenEval, the T2I-CompBench++, the GenAI-Bench, and the DPG-Bench, respectively. For the five scenarios, each involving the removal of one of the five components from Diffusion-RainbowPA (the label "w/o X" indicates the alignment performance without the component "X"). Firstly, observing all test results across the four datasets, it can be found that each component has positively contributed to the alignment performance of Diffusion-RainbowPA. For the observed bias in GenEval, it primarily stems from GenEval's nature of short prompts and a relatively small prompt set (comprising only 553 prompts). Of the three remaining datasets, the model excluding JS

Table 5: Ablation Study on the GenEval.

| GenEval | VQAS ↑ | CLIPS ↑ | HPS ↑ | IR ↑ |
|---------|--------|---------|-------|------|
| SD-1.5 | 61.85 | 33.77 | 26.93 | -0.200 |
| Ours | 68.41 | 35.01 | 27.45 | 0.202 |
| w/o CEPA | 68.05 | 34.77 | 27.49 | **0.239** |
| w/o IPA | 68.42 | 34.81 | 27.50 | 0.218 |
| w/o JS | 68.14 | 34.95 | **27.54** | 0.210 |
| w/o MSPA | **68.94** | **35.04** | 27.47 | 0.224 |
| w/o SFT-L | 67.25 | 34.43 | 27.45 | 0.197 |

Table 6: Ablation Study on the T2I-CompBench++.

| T2I-Comp++ | VQAS ↑ | CLIPS ↑ | HPS ↑ | IR ↑ |
|------------|--------|---------|-------|------|
| SD-1.5 | 60.71 | 32.06 | 26.55 | -0.245 |
| Ours | **67.60** | **33.48** | 27.15 | **0.230** |
| w/o CEPA | 66.84 | 33.14 | 27.18 | 0.203 |
| w/o IPA | 66.55 | 33.23 | 27.17 | 0.172 |
| w/o JS | 67.29 | 33.36 | **27.23** | 0.219 |
| w/o MSPA | 67.16 | 33.20 | 27.12 | 0.184 |
| w/o SFT-L | 67.48 | 33.06 | 27.13 | 0.180 |

Table 7: Ablation Study on the GenAI-Bench.

| GenAI-Bench | VQAS ↑ | CLIPS ↑ | HPS ↑ | IR ↑ |
|-------------|--------|---------|-------|------|
| SD-1.5 | 59.50 | 33.62 | 26.94 | 0.237 |
| Ours | 62.58 | **34.47** | 27.53 | **0.593** |
| w/o CEPA | 62.23 | 34.12 | 27.57 | 0.531 |
| w/o IPA | 61.55 | 34.25 | 27.54 | 0.517 |
| w/o JS | 62.24 | 34.16 | **27.59** | 0.542 |
| w/o MSPA | **62.78** | 34.04 | 27.51 | 0.505 |
| w/o SFT-L | 62.24 | 34.01 | 27.53 | 0.485 |

Table 8: Ablation Study on the DPG-Bench.

| DPG-Bench | VQAS ↑ | CLIPS ↑ | HPS ↑ | IR ↑ |
|-----------|--------|---------|-------|------|
| SD-1.5 | 76.28 | 33.42 | 26.31 | -0.187 |
| Ours | **79.18** | **34.72** | 26.95 | **0.166** |
| w/o CEPA | 78.73 | 34.38 | 26.95 | 0.088 |
| w/o IPA | 78.30 | 34.57 | 26.95 | 0.075 |
| w/o JS | 78.79 | 34.36 | **26.97** | 0.111 |
| w/o MSPA | 78.78 | 34.49 | 26.93 | 0.054 |
| w/o SFT-L | 78.56 | 34.46 | 26.94 | 0.023 |

demonstrated superior performance on HPS-V2 compared to the full model. Such observation is related to a potential conflict between the optimization objective of JS divergence and the evaluation dimensions of HPS-V2: JS divergence aims to stabilize the training process and alleviate overfitting by symmetrizing KL divergence. Its core objective is to improve the distribution consistency of generated samples, rather than directly optimizing the fine-grained attributes of human preferences (such as color accuracy, reasonable composition); then the distribution over-smoothing and optimization direction shift introduced by the JS divergence paradigm might slightly decrease the HPS-V2 results. Despite this, performance with the JS divergence always holds significant advantages over that with the KL divergence in all other metrics: for example, as shown in Table 5, ablating the JS results in a 1.05 decrease in VQAScore and a 0.06 decrease in CLIPScore; in Table 6, removing the JS leads to reductions of 0.31 in VQAScore, 0.12 in CLIPScore, and 0.011 (5%) in ImageReward; in Table 7, JS removal decreases VQAScore by 0.34, CLIPScore by 0.31, and ImageReward by 0.051 (9%); in Table 8, removing JS lowers VQAScore by 0.39, CLIPScore by 0.36, and ImageReward by 0.143 (86%). Hence, considering all test results comprehensively, we can conclude that omission of any single component would result in a degradation of the alignment performance: that is to say, regarding the introduced five components from three aspects as *"the whole"* can get *"greater"* performance than that of *"its parts"*.

# 6 Conclusion

In this study, we point out limitations of current state-of-the-art diffusion-based text-to-image alignment methods, which tend to suffer from text-image misalignment, aesthetic overfitting and low-quality generation. To address such issues, we improve the alignment paradigm through a tripartite perspective: calibration enhancement, overfitting mitigation and performance optimization. For calibration enhancement, we introduce the Calibration Enhancement Preference Alignment (CEPA); for overfitting mitigation, we introduce the Identical Preference Alignment (IPA) and the Jensen-Shannon Divergence Constraint; for performance optimization, we introduce the Margin Strengthened Preference Alignment (MSPA) and the SFT-like Regularization. Further combining the introduced five components with step-aware preference alignment paradigm, we propose the Diffusion-RainbowPA, a suite of six improvements that collectively enhance the alignment performance of

Diffusion-DPO. Comprehensive evaluations demonstrate that Diffusion-RainbowPA outperforms current state-of-the-art methods. Furthermore, ablation study conducted on the five introduced components indicates that each component of Diffusion-RainbowPA has positively contributed to the alignment performance.

## Broader Impact Statement

Performance of Diffusion-RainbowPA is promising, while actually any effort in text-to-image generation presents ethical risks. The improved models might be misused to generate harmful, hateful, fake or sexually explicit content. We further utilize supplementary safety filtering mechanisms during inference to ascertain the elimination of toxic content. Specifically, we train only the UNet model and do not train the safety checker model to prevent potential vulnerabilities, such as the reward hacking, that could attack the safety checker model during the training process. Moreover, as with all preference-based alignment methods, the biases of preference that has been encoded in the dataset might be introduced. For this problem, we adopt the solution consistent with previous methods like Diffusion-DPO and SPO, i.e. performing data filtering; and we utilize the same datasets with them in our training process.

## Acknowledgments

This work was supported by the Natural Science Foundation of Shenzhen (No.JCYJ20230807111604008, No.JCYJ20240813112007010), the Natural Science Foundation of Guangdong Province (No.2024A1515010003), National Key Research and Development Program of China (No.2022YFB4701400) and Cross-disciplinary Fund for Research and Innovation (No.JC2024002) of Tsinghua SIGS.

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

# A    Diffusion-based Text-to-Image Models

Early researches in text-to-image (T2I) generation mainly focus on GANs (Reed et al., 2016; Zhang et al., 2017). More recently, however, diffusion-based models have demonstrated remarkable proficiency in producing images of exceptional quality, adhering to the input descriptions better (Ho et al., 2020; Rombach et al., 2022; Podell et al., 2024; Esser et al., 2024; Ramesh et al., 2021; 2022; Betker et al., 2023; Yang et al., 2023). To achieve more precise control over the generative process, recent studies have explored various techniques aimed at enhancing the guidance of diffusion models. DiT (Peebles & Xie, 2023) exemplifies better generation with utilization of superior transformer backbones; Pixart-alpha (Chen et al., 2024) exemplifies efficacy of advanced text encoders in the accomplishment of robust language conditioning; DALL·E 3 (Betker et al., 2023) exemplifies enhanced generative performance through improved captioning. Despite their effectiveness, aligning diffusion models with human value still presents a significant challenge and remains a critical issue. In this study, we utilize the Stable Diffusion v1.5 (Rombach et al., 2022) as our backbone, primarily based on the goal of aligning with the RL-based T2I alignment research group. Herein, we would like to briefly introduce the core of diffusion-based T2I for readers who are not very familiar with it, which is the mechanism of Denoising Diffusion Probabilistic Models (DDPMs).

Let's consider the data distribution $x_0 \sim q_0(x_0), x_0 \in \mathbb{R}^n$. DDPM algorithm approximates the data distribution $q_0$ with a parameterized model with the form of $p_\theta(x_0) = \int p_\theta(x_{0:T}|c)dx_{1:T}$, where $p_\theta(x_{0:T}|c) = p_\theta(x_T) \prod_{t=1}^{T} p_\theta(x_{t-1}|x_t, c)$, and $c$ is the conditioning information, (i.e., the image category and the image caption). Then, we can describe *the reverse process* to be an Markov chain with dynamics as follows:

$$p(x_T) = \mathcal{N}(0, I), p_\theta(x_{t-1}|x_t, c) = \mathcal{N}(x_{t-1}; \mu_\theta(x_t, c), \Sigma_t).$$

Furthermore, DDPM further exploits an approximate posterior $q(x_{1:T}|x_0, c)$, namely *the forward process*, adding Gaussian noise to the data acccording to the variance coefficients $\beta_1, ..., \beta_T$:

$$q(x_{1:T}|x_0, c) = \prod_{t=1}^{T} q(x_t|x_{t-1}, c),$$

$$q(x_t|x_{t-1}, c) = \mathcal{N}(\sqrt{1 - \beta_t}x_{t-1}, \beta_t I),$$

$$\alpha_t = 1 - \beta_t, \widetilde{\alpha}_t = \prod_{i=1}^{t} \alpha_i, \widetilde{\beta}_t = \frac{1 - \widetilde{\alpha}_{t-1}}{1 - \widetilde{\alpha}_t}.$$

Based on these, in the work (Ho et al., 2020), the parameterization is further applied as follows:

$$\mu_\theta(x_t, c) = \frac{1}{\sqrt{\alpha_t}}(x_t - \frac{\beta_t}{\sqrt{1 - \widetilde{\alpha}_t}}\epsilon_\theta(x_t, c))$$

# B    Brief Survey of Text-to-Image Alignment

Reinforcement learning (RL) has recently exhibited significant potential for application across a diverse array of domains (Sun et al., 2025a; Xia et al., 2025). Furthermore, in the domain of aligning diffusion-based text-to-image models with human preferences, a prevalent method family is the Reinforcement Learning from Human Feedback (RLHF). To summarize, they encompass the maximization of target reward function in conjunction with the minimization of Kullback-Leibler divergence between current policy and reference policy. Methods such as DDPO (Black et al., 2024), DPOK (Fan et al., 2024), ReFL (Xu et al., 2024), DRaFT (Clark et al., 2024), and AlignProp (Prabhudesai et al., 2023) initially entail training a reward model for modeling human value, followed by employing RL pipelines like PPO (Schulman et al., 2017) and REINFORCE (Sutton et al., 1999) to fine-tune the policy, thereby optimizing for the rewards offered by the reward model. Despite their effectiveness, traditional RLHF methods involve the modeling of reward from relatively limited preference datasets, which results in the training pipeline suffer from disadvantages such as high computational costs and inadequate training stability.

Direct Preference Optimization (DPO) (Rafailov et al., 2024) achieves remarkable success in Large Language Models (LLMs) alignment by implicitly estimating the reward model. Based on this, some prior work has been conducted grounding in the idea of implicit reward models (Liu et al., 2024a). Diffusion-DPO (Wallace et al., 2024) re-formulates DPO by utilizing the evidence lower bound (ELBO) to derive a differentiable objective function, and further approximation the reverse process with the forward. D3PO (Yang et al., 2024a) regards the denoising process as a multi-step Markov decision process (MDP), drawing the conclusion that directly updating the policy based on human preferences within MDP is equivalent to first learning the optimal reward function. DenseReward (Yang et al., 2024b) further advances the DPO paradigm by introducing a temporal discounting, which accentuates the initial stages of denoising while comparatively weaken posterior ones. Recently, SPO (Liang et al., 2024) is proposed to evaluate and adjust the denoising performance at step-level for ensuring accurate step-specific preference signal. These paradigms have laid a robust groundwork for researchers to pursue further advancements.

Building upon them, recent researches on aligning diffusion-based text-to-image with human preference primarily focus on three directions. The initial aspect involves deriving inspiration from existing numerous alignment paradigms of generative models (such as LLMs), exploring and modifying them for application in the context of diffusion-based text-to-image. The second aspect is to further integrate DPO paradigm with diffusion model itself for further leveraging the advantages of denoising process. The third aspect involves exploration of novel training paradigms to achieve superior performance and better efficiency during the training process.

In the first aspect aforementioned, some typical works are as follows. Diffusion-KTO (Li et al., 2024b) formulates the alignment objective as the maximization of expected human utility and further offers a robust framework based on existing KTO methodology. Diffusion-RPO (Gu et al., 2024) applies contrastive weighting to similar prompt-image pairs based on the success of RPO. MaPO (Hong et al., 2024) eschews conventional reference paradigm, concurrently optimizing the likelihood margin between the preferred and the dispreferred image sets and augmenting the likelihood of preferred set, which is also partly inspired by SimPO. Overall, there still exists substantial potential for further investigation within this aspect.

In regard to the second aforementioned aspect, some representative works are delineated as follows. In the studies of DNO (Tang et al., 2024), NCPPO (Gambashidze et al., 2024), and ReNO (Eyring et al., 2024), researchers delve into examination of the impact of noise on resultant outcomes, employing methodologies primarily centered around the noise optimization. DAS (Kim et al., 2025) offers a training-free approach to aligning diffusion models with arbitrary reward functions, thereby maintaining their generalizable performance.

Concerning the third aspect, several exemplary works are discussed as follows. SPIN-Diffusion (Yuan et al., 2024) enables the diffusion model to compete with its earlier versions, thereby facilitating an iterative process of self-improvement. DUO (Park et al., 2024) endeavors to eliminate Not Safe For Work (NSFW) content from T2I models, simultaneously safeguarding their proficiency across unrelated subjects. SafetyDPO (Liu et al., 2024b) facilitates safety alignment of T2I models, precluding the generation of inappropriate outputs without impacting generative capabilities on safe prompts. PopAlign (Li et al., 2024c) proposes a concept of population-level preference optimization as a means of mitigating the population bias. RankDPO (Karthik et al., 2024) weighs the preference loss with discounted cumulative gains, which further enhances DPO-based methods with the help of ranking feedback. SePPO (Zhang et al., 2024) further employs previously saved checkpoints as reference models, concurrently utilizing them to produce on-policy samples. SEE-DPO (Shekhar et al., 2024) incorporates a self-entropy regularization term into conventional KL-regularized formulation of RLHF objective function. In the study (Sun et al., 2025b), it is proposed that employment of Jensen–Shannon divergence yields superior performance in achieving human value alignment comparing to Kullback-Leibler divergence, while concurrently facilitating an optimal trade-off between alignment performance and generative diversity. PatchDPO (Huang et al., 2024) estimates the quality of image patches within each generated image and accordingly trains the model.

In summary, exploration on aligning diffusion-based text-to-image models with human preferences is yet to traverse a substantial expanse; concurrently, this field is undergoing a burgeoning phase. The aforementioned three aspects, along with their interwoven integration, constitute the predominant pathways for future researches.

# C   More Details on the Evaluation and Comparison

In this part, we provide more detailed descriptions on the evaluation metrics, evaluation datasets and comparison methods.

## C.1   Details on the Metrics

In this study, we utilize total four published zero-shot metrics: the VQAScore (Lin et al., 2025), the CLIPScore (Radford et al., 2021), the HPS-V2 (Wu et al., 2023), and the ImageReward (Xu et al., 2024).

**VQAScore.** VQAScore utilizes the Visual Question Answering (VQA) model for evaluation. Its core idea is to transform the given text into a simple yes/no question: "Does this image depict text?" The image and the question are then input into a pre-trained VQA model, and the probability of the model predicting "Yes" is used as the matching score between the image and the text. To further enhance the performance of VQAScore, the authors also trained an internal VQA model named *CLIP-FlanT5*. This model employs a bidirectional image-question encoder, allowing image embeddings to be adjusted according to the question and vice versa, which is more in line with how humans understand images. CLIP-FlanT5 has achieved a new state-of-the-art level in all benchmark tests. The simplicity and effectiveness of VQAScore make it a powerful tool for evaluating text-to-visual generation models, particularly demonstrating significant advantages in handling complex text prompts.

**CLIPScore.** CLIPScore is a widely used metric for evaluating the quality of generated images, which leverages OpenAI's CLIP (Contrastive Language-Image Pre-training) model to measure the semantic consistency between a generated image and a given text description. The core idea is: first, use the CLIP model to separately compute the embedding vectors for the generated image and the text description; then, calculate the *cosine similarity* between these two vectors. The higher the similarity score, the higher the semantic alignment between the image and the text, indicating that the image better expresses the content of the text description. Advantage of CLIPScore lies in its strong generalization ability and attention to image details, as it is trained on large-scale image-text datasets, capable of capturing richer semantic information, and not just focusing on the visual quality of the image, but also on its alignment with the text description. Although CLIPScore can to some extent reflect human perception of image quality and text consistency, it is not perfect and may be influenced by certain specific text prompts, and there may be limitations in understanding complex scenes and abstract concepts. Nevertheless, CLIPScore remains an important tool for evaluating the performance of generation models, often used in combination with other metrics to more comprehensively assess the quality of generated images.

**HPS-V2.** HPS-V2 is trained on a large human preference dataset, the Human Preference Dataset v2 (HPD-V2), which contains 798,090 human preference choices on 433,760 pairs of images, making it the largest dataset of its kind. To avoid biases present in previous datasets (such as primarily containing images generated by specific models or using biased prompts), HPD-V2 deliberately collects images from 9 different text-to-image generation models (including Stable Diffusion, DALL-E 2, etc.) and the COCO dataset. It also uses ChatGPT to clean up the prompts, making them more concise and clear, reducing stylistic language and contradictory information. HPS-V2 is obtained by fine-tuning the CLIP model on HPD-V2. Experiments show that it has better generalization capabilities than previous metrics (such as HPS-V1, PickScore, etc.), can better predict human preferences for generated images.

**ImageReward.** ImageReward is a general human preference scoring model. It is trained through a systematic annotation process that includes two stages: scoring and ranking, where over 1.37 million expert comparison data has been collected. The process first classifies and identifies issues in the text prompts, then scores generated images on three dimensions: text-image alignment, fidelity, and harmlessness, and finally ranks the images to capture human preferences. Experiments show that ImageReward outperforms existing scoring models in understanding human preferences, such as CLIP, Aesthetic, and BLIP. In evaluations of real user prompts and the MS-COCO 2014 dataset, ImageReward is highly consistent with human preference rankings and has a higher degree of discrimination between samples, making it a promising automatic evaluation metric for text-to-image generation models.

## C.2 Details on the Datasets

In evaluations of this study, we utilize four published benchmarking datasets. Based on nature of prompts in the dataset, we can categorize them into **the short prompts** and **the long prompts**. Herein, the short prompts include the GenEval (Ghosh et al., 2024) and the T2I-CompBench++ (Huang et al., 2025); the long prompts include the GenAI-Bench (Li et al., 2024a) and the DPG-Bench (Hu et al., 2024).

**GenEval.** GenEval is a relatively small dataset (containing only 553 prompts) designed for evaluating the generation capabilities of text-to-image (T2I) models. GenEval decomposes text prompts into their constituent parts, such as number, attribution, color, and relative position. It then uses an object detection model to identify objects in the generated images, extracting bounding box and segmentation mask information. This information is used to verify the presence of specified objects, the accuracy of their counts, and the adherence to the described spatial relationships between objects. In this research, we do not use the benchmarking results from the GenEval dataset itself but instead use the quantitative results from metrics in Appendix C.1 with the purpose of unifying results' comparison across different datasets.

**T2I-CompBench++.** T2I-CompBench++ is a benchmark comprised of 8000 synthetic text prompts designed to evaluate the compositional capabilities of text-to-image generation models. It categorizes prompts into four main areas: Attribute Binding, Object Relationships, Generation Arithmetic, and Complex Combinations, further subdividing them into eight subcategories, including newly introduced 3D spatial relationships and numeracy. Attribute Binding comprises subcategories for Color, Shape, and Texture, assessing the model's ability to associate attributes with the correct objects. Object Relationships includes subcategories for 2D/3D Spatial Relationships and Non-Spatial Relationships, evaluating the model's understanding and generation of various object interactions. Generation Numeracy assesses the model's capability to handle text prompts specifying different object quantities. Complex Combinations involve combining multiple objects or categories, testing the model's ability to handle more complex scenarios. To evaluate these diverse compositional challenges, T2I-CompBench++ proposes improved evaluation metrics, including detection-based metrics for 3D spatial relationships and arithmetic, as well as analytical metrics leveraging multimodal large language models (MLLMs) such as GPT-4V. By benchmarking 11 text-to-image models, including FLUX.1, SD3, DALL-E 3, Pixart-$\alpha$, and SDXL, on T2I-CompBench++, and conducting thorough evaluations to validate the metrics' effectiveness, as well as exploring the potential and limitations of MLLMs, T2I-CompBench++ provides a robust framework for assessing and advancing compositional capabilities in text-to-image generation. Similarly, we do not use the benchmarking results itself but instead use the quantitative results from metrics in Appendix C.1 with the purpose of unifying results' comparison across all different datasets.

**GenAI-Bench.** GenAI-Bench is a novel benchmark designed for comprehensive evaluation of compositional text-to-visual generation models. Unlike previous benchmarks that primarily focus on basic visual elements like objects, attributes, and simple relationships, GenAI-Bench incorporates a wider range of compositional skills, categorized into "basic" (objects, attributes, spatial/action/part relations, scenes) and "advanced" (counting, comparison, differentiation, negation, universality) skills. This expanded skill taxonomy reflects the complexities of real-world user prompts, sourced directly from professional designers familiar with tools like Midjourney. The benchmark includes 1600 challenging prompts, each meticulously tagged with all relevant skills, enabling fine-grained analysis of model performance across different aspects of compositional reasoning. The study utilizes over 80,000 human ratings (38,400 for initial model evaluation, and an additional 43,200 for a GenAI-Rank sub-benchmark focused on image ranking) on images and videos generated by ten leading models (including both open-source and closed-source options), revealing significant shortcomings in handling advanced reasoning tasks even for state-of-the-art models. Furthermore, GenAI-Bench facilitates the evaluation of automated metrics, demonstrating the superior performance of VQAScore – a metric leveraging VQA models – compared to existing methods like CLIPScore, especially in correlating with human judgments on compositional prompts. Similarly, we do not use the VQAScore merely but use the four quantitative results from metrics in Appendix C.1 with the purpose of comprehensively carrying out the comparison.

**DPG-Bench.** DPG-Bench is a dataset introduced to evaluate ability of text-to-image models to handle complex and dense prompts. Previous benchmarks like T2I-CompBench and PartiPrompts primarily focus on shorter prompts with limited descriptive detail. DPG-Bench addresses this limitation by providing a more challenging evaluation set. It consists of 1,065 lengthy and detailed prompts, significantly exceeding the length

and complexity of those in previous benchmarks. These prompts describe multiple objects, each with diverse attributes and intricate relationships between them. The average prompt in DPG-Bench contains around 84 tokens, compared to 10-20 tokens in previous benchmarks. The increase reflects a much richer semantic content within each prompt. Source data is gathered from COCO, PartiPrompts, DSG-1k, and Object365. And the existing short prompts are extended using GPT-4 to incorporate detailed descriptions of objects, attributes, and relationships. They are then verified by human annotators to ensure quality and accuracy. Similarly, we use the quantitative results from metrics in Appendix C.1 with the purpose of comprehensively carrying out the comparison.

### C.3 Details on the Comparison Methods

In the comparison of this paper, we mainly focus on the Diffusion-DPO, the SPO (with and without LoRA), the SPIN-Diffusion, and the SePPO.

**Diffusion-DPO. (CVPR 2024)** Diffusion-DPO is the first attempt designed to align text-to-image diffusion models with human preferences. It departs from traditional approaches using reinforcement learning or specific datasets by directly optimizing the diffusion model. Instead of crafting an explicit reward function, DPO subtly learns human preferences by directly tweaking the model's parameters. It utilizes paired image comparisons where one image is preferred over another for a given prompt. The aim is to improve the win ratio of preferred images relative to dispreferred ones under the model's distribution.

**SPO. (CVPR 2025)** Step-by-Step Preference Optimization (SPO) (with LoRA and without LoRA) is a novel post-training method for enhancing the aesthetic quality of diffusion models. Unlike existing DPO-based methods that propagate preferences across entire generation trajectories, SPO focuses on fine-grained detail improvements at each denoising step. SPO operates by first sampling a pool of candidate images from a shared noisy latent at each step. A Step-aware Preference Model (SPM), trained on open-source preference data, then evaluates these candidates and identifies a win-lose pair exhibiting the largest quality difference: crucially, these pairs are visually similar, highlighting subtle aesthetic details rather than large layout discrepancies. The diffusion model is then fine-tuned using a modified DPO loss function based on this win-lose pair. Finally, a candidate image is randomly selected from the pool to initialize the next denoising step. Such iterative process, repeated across multiple steps, accumulates minor aesthetic improvements, leading to significantly enhanced overall visual appeal. Key advantage of SPO is its ability to leverage generic preference data effectively by focusing on small, detail-level differences at each step, circumventing the limitations of existing DPO methods that struggle with noisy and conflicting preferences in holistic image evaluations.

**SPIN-Diffusion. (NeurIPS 2024)** SPIN-Diffusion is a novel self-play fine-tuning method for diffusion models, addressing the limitations of standard supervised fine-tuning (SFT) and reinforcement learning from human feedback (RLHF). Unlike SFT, which plateaus with limited data and doesn't directly optimize for human preference; and unlike RLHF, which requires paired "winner-loser" images per prompt, SPIN-Diffusion iteratively improves a diffusion model by pitting it against previous versions. Core idea of SPIN-Diffusion is a minimax game: a "main player" (the current model) tries to distinguish between real images from the target distribution and those generated by an "opponent player" (a previous model version). The opponent aims to fool the main player. Crucially, the opponent is simply a copy of the main player from a previous iteration, enabling self-play. SPIN-Diffusion overcomes challenges inherent in applying self-play to diffusion models by (a) designing an objective function considering the entire diffusion trajectory (not just the final image), and (b) decomposing and approximating the probability function using score functions (gradients of probabilities), leveraging the Gaussian reparameterization technique from DDIM for efficient computation. Hence, it allows for an unbiased objective function calculated from intermediate samples, further approximated for computational efficiency to avoid storing all intermediate images. The iterative self-play process continues until convergence, theoretically achieving a point where further SFT improvement is impossible.

**SePPO. (arXiv preprint)** Semi-Policy Preference Optimization (SePPO) is a novel method for aligning diffusion models (DMs) with human preferences without relying on reward models or paired human-annotated data. It addresses limitations of both on-policy and off-policy reinforcement learning from human feedback (RLHF) approaches. SePPO leverages previous DM checkpoints as reference models. They generate on-policy reference samples, acting as replacements for "losing images" in preference pairs. It further optimizes using

only off-policy "winning images" from a supervised fine-tuning (SFT) dataset. A key innovation is the reference model selection strategy. Instead of using initial or latest checkpoint, SePPO randomly samples from all previous checkpoints, broadening exploration in policy space and thus preventing overfitting. Furthermore, SePPO employs an Anchor-based Adaptive Flipper (AAF). AAF assesses whether reference samples are truly inferior to current model's output using a criterion based on the probability of generating the winning image. This adaptive mechanism prevents performance degradation due to uncertain reference sample quality, selectively learning from generated samples based on their likely classification as winning or losing.

## D   More Detailed Results of Comparison with SOTAs on the T2I-CompBench++ with Eight Categories Separately

Table 9: Comparison of Diffusion-RainbowPA with SOTAs on the T2I-CompBench++ (eight categories separately).

| VQAScore | | | | | | | | |
|---|---|---|---|---|---|---|---|---|
| **T2I-Comp++** | 3d_spatial | color | complex | non-spatial | numeracy | shape | spatial | texture |
| SD-1.5 | 50.95 | 55.82 | 70.50 | 76.37 | 49.52 | 59.72 | 61.07 | 61.70 |
| Diffusion-DPO | 54.26 | 57.94 | 72.11 | 77.67 | 50.46 | 61.06 | 64.47 | 62.95 |
| SPO | 57.57 | 59.01 | 72.83 | 72.96 | 50.32 | 63.67 | 63.07 | 62.44 |
| SPO (LoRA) | 57.63 | 59.20 | 72.91 | 73.00 | 50.40 | 63.74 | 63.00 | 62.31 |
| SPIN-Diffusion | 49.96 | 57.01 | 71.85 | 74.65 | 47.17 | 62.47 | 62.48 | 61.15 |
| SePPO | 54.42 | 60.68 | 72.56 | 78.09 | 50.26 | 60.70 | 65.52 | 62.54 |
| **Ours** | **61.69** | **66.23** | **74.48** | **79.16** | **53.64** | **65.63** | **69.06** | **70.89** |
| CLIPScore | | | | | | | | |
| SD-1.5 | 31.46 | 32.76 | 30.98 | 33.19 | 31.54 | 30.94 | 33.91 | 31.67 |
| Diffusion-DPO | 32.26 | 33.60 | 31.31 | 33.61 | 32.22 | 31.48 | 34.78 | 32.03 |
| SPO | 31.33 | 32.21 | 29.87 | 31.15 | 30.80 | 30.68 | 33.07 | 30.38 |
| SPO (LoRA) | 31.32 | 32.21 | 29.87 | 31.14 | 30.79 | 30.68 | 33.10 | 30.36 |
| SPIN-Diffusion | 30.58 | 31.85 | 29.67 | 32.65 | 30.53 | 30.79 | 33.56 | 29.61 |
| SePPO | 31.63 | 33.40 | 31.07 | 33.36 | 32.36 | 31.11 | 34.18 | 31.31 |
| **Ours** | **33.45** | **34.58** | **31.53** | **33.68** | **33.37** | **32.12** | **35.66** | **33.46** |
| HPS-V2 | | | | | | | | |
| SD-1.5 | 26.87 | 27.06 | 25.61 | 26.62 | 26.17 | 26.27 | 27.91 | 25.91 |
| Diffusion-DPO | 27.12 | 27.29 | 25.74 | 26.80 | 26.39 | 26.50 | 28.29 | 26.10 |
| SPO | 27.21 | 27.44 | 25.99 | 26.81 | 26.37 | 26.61 | 28.42 | 26.15 |
| SPO (LoRA) | 27.21 | 27.44 | 25.99 | 26.81 | 26.37 | 26.61 | 28.42 | 26.15 |
| SPIN-Diffusion | 27.38 | 27.52 | 25.99 | 27.05 | 26.47 | 26.69 | 28.62 | 26.25 |
| SePPO | 27.39 | 27.64 | 26.03 | 27.10 | 26.65 | 26.63 | 28.55 | 26.27 |
| **Ours** | **27.54** | **27.71** | **26.06** | **27.13** | **26.77** | **26.76** | **28.68** | **26.51** |
| ImageReward | | | | | | | | |
| SD-1.5 | -0.552 | -0.496 | -0.092 | 0.549 | -0.500 | -0.431 | -0.043 | -0.393 |
| Diffusion-DPO | -0.341 | -0.322 | 0.030 | 0.670 | -0.399 | -0.320 | 0.244 | -0.275 |
| SPO | -0.113 | -0.331 | 0.116 | 0.601 | -0.228 | -0.163 | 0.285 | -0.352 |
| SPO (LoRA) | -0.108 | -0.327 | 0.116 | 0.606 | -0.228 | -0.164 | 0.293 | -0.359 |
| SPIN-Diffusion | -0.329 | -0.359 | 0.121 | 0.723 | -0.452 | -0.176 | 0.233 | -0.446 |
| SePPO | -0.246 | -0.179 | 0.157 | 0.805 | -0.295 | -0.281 | 0.313 | -0.321 |
| **Ours** | **0.059** | **0.106** | **0.200** | **0.819** | **-0.051** | **-0.017** | **0.553** | **0.168** |

# E More Detailed Results of Ablation Study on the T2I-CompBench++ with Eight Categories Separately

Table 10: Ablation study of Diffusion-RainbowPA on the T2I-CompBench++ (eight categories separately).

| VQAScore | | | | | | | | |
|---|---|---|---|---|---|---|---|---|
| **T2I-Comp++** | 3d_spatial | color | complex | non-spatial | numeracy | shape | spatial | texture |
| SD-1.5 | 50.95 | 55.82 | 70.50 | 76.37 | 49.52 | 59.72 | 61.07 | 61.70 |
| Ours | 61.69 | 66.23 | 74.48 | 79.16 | 53.64 | 65.63 | 69.06 | 70.89 |
| w/o CEPA | 60.96 | 64.74 | 74.95 | 78.73 | 52.85 | 64.54 | 68.57 | 69.37 |
| w/o IPA | 60.24 | 64.76 | 74.34 | 79.09 | 52.96 | 64.39 | 67.83 | 68.80 |
| w/o JS | 61.64 | 64.57 | 74.74 | 79.08 | 54.07 | 64.66 | 69.89 | 69.62 |
| w/o MSPA | 60.57 | 65.30 | 74.60 | 78.62 | 53.89 | 64.43 | 69.19 | 70.69 |
| w/o SFT-L | 61.07 | 66.76 | 74.55 | 79.46 | 53.78 | 64.91 | 69.13 | 70.16 |
| **CLIPScore** | | | | | | | | |
| SD-1.5 | 31.46 | 32.76 | 30.98 | 33.19 | 31.54 | 30.94 | 33.91 | 31.67 |
| Ours | 33.45 | 34.58 | 31.53 | 33.68 | 33.37 | 32.12 | 35.66 | 33.46 |
| w/o CEPA | 33.10 | 34.32 | 31.63 | 33.39 | 32.91 | 31.65 | 35.18 | 32.95 |
| w/o IPA | 33.13 | 34.48 | 31.72 | 33.48 | 32.92 | 31.98 | 35.31 | 32.83 |
| w/o JS | 33.22 | 34.36 | 31.51 | 33.51 | 33.45 | 32.00 | 35.61 | 33.25 |
| w/o MSPA | 32.94 | 34.22 | 31.47 | 33.44 | 32.85 | 31.80 | 35.33 | 33.58 |
| w/o SFT-L | 32.76 | 34.21 | 31.50 | 33.05 | 32.84 | 31.65 | 35.01 | 33.46 |
| **HPS-V2** | | | | | | | | |
| SD-1.5 | 26.87 | 27.06 | 25.61 | 26.62 | 26.17 | 26.27 | 27.91 | 25.91 |
| Ours | 27.54 | 27.71 | 26.06 | 27.13 | 26.77 | 26.76 | 28.68 | 26.51 |
| w/o CEPA | 27.53 | 27.75 | 26.16 | 27.24 | 26.77 | 26.77 | 28.72 | 26.54 |
| w/o IPA | 27.56 | 27.76 | 26.15 | 27.16 | 26.76 | 26.81 | 28.70 | 26.48 |
| w/o JS | 27.65 | 27.80 | 26.14 | 27.21 | 26.89 | 26.84 | 28.82 | 26.54 |
| w/o MSPA | 27.46 | 27.70 | 26.09 | 27.12 | 26.75 | 26.77 | 28.62 | 26.51 |
| w/o SFT-L | 27.50 | 27.78 | 26.11 | 27.14 | 26.73 | 26.76 | 28.61 | 26.48 |
| **ImageReward** | | | | | | | | |
| SD-1.5 | -0.552 | -0.496 | -0.092 | 0.549 | -0.500 | -0.431 | -0.043 | -0.393 |
| Ours | 0.059 | 0.106 | 0.200 | 0.819 | -0.051 | -0.017 | 0.553 | 0.168 |
| w/o CEPA | 0.018 | 0.073 | 0.243 | 0.796 | -0.057 | -0.097 | 0.543 | 0.104 |
| w/o IPA | -0.010 | 0.028 | 0.206 | 0.759 | -0.109 | -0.066 | 0.518 | 0.047 |
| w/o JS | 0.079 | 0.069 | 0.194 | 0.791 | -0.017 | -0.067 | 0.583 | 0.121 |
| w/o MSPA | -0.025 | 0.035 | 0.186 | 0.765 | -0.108 | -0.066 | 0.513 | 0.175 |
| w/o SFT-L | 0.025 | 0.041 | 0.137 | 0.759 | -0.134 | -0.095 | 0.550 | 0.158 |

# F Further Discussion on Computational Overhead and Complexity Introduced

In this study, the experiments are conducted on a machine equipped with 4 × NVIDIA A100-PCIE-40GB GPUs. The training process for Diffusion-RainbowPA requires approximately 15 hours and the maximum GPU memory consumption required by Diffusion-RainbowPA is 21.74 GB per GPU, indicating that it is a relatively lightweight method. Furthermore, we perform experiments on consumer-grade graphics cards, specifically utilizing a machine equipped with 4 × NVIDIA GeForce RTX 3090 GPUs (each with 24GB of memory); and the entire training process on this setup requires approximately 27.5 hours.

In addition, we provide an intuitive explanation of the complexity introduced by integrating multiple alignment components. The six alignment improvements in Diffusion-RainbowPA are implemented as additional regularization terms within a similar training loop to that of the baseline SPO (Liang et al., 2024). Herein, in contrast to SPO, which employs a multi-sample setup in the step-wise resampler by sampling four images, we instead adopt a configuration that samples only two images. This modification partially reduces computational complexity. For the CEPA, two additional squared error terms are introduced, which increase only scalar multiplication, addition, and squaring operations for the computation; hence, CEPA can be considered a quadratic regularization extension of $\log \frac{p_\theta(x_{t-1}|c,t,x_t)}{p_{\text{ref}}(x_{t-1}|c,t,x_t)}$, without significantly increasing algorithmic complexity. Similarly, IPA only adds scalar subtraction and squaring operations; JS introduces only scalar multiplication, addition, and division; MSPA adds only scalar multiplication and addition; and SFT-L requires only comparison and truncation operations. Hence, none of them significantly increase the algorithmic complexity. Overall, the added complexity resulting from the integration of multiple alignment components is deemed acceptable.

## G  Further Discussion on Hyperparameter Tuning

In this study, hyperparameter tuning is primarily guided by two main principles. Firstly, we employ a unified initialization for the regularization intensity $\beta$, assigning the same value to all of them. For a fair comparison, we set this value to 10, consistent with the configuration used in D3PO (Yang et al., 2024a) and SPO (Liang et al., 2024). Secondly, we empirically anchor each component by fixing their values based on results from preliminary experiments. Specifically, we perform an individual grid search for each hyperparameter, including the SFT intensity $\lambda$, the threshold $\theta$, and the margin strengthening intensity $\eta$. While this might not be the globally optimal combination of hyperparameters, we want to convey that this paper transcends the pursuit of immediate performance enhancements and its contribution more lies in the articulation of novel research avenues within the field, facilitated by methodological innovation.

In future engineering applications, especially when handling high-dimensional data or large-scale models, hyperparameter optimization can be achieved through the following method. To begin, we need to define the acceptable range of values for each hyperparameter; and if there are sufficient resources, we can set it as a continuous range for finer-grained tuning, while, if resources are limited, we can empirically set it as discrete values to reduce the search space. For example, for the regularization intensity $\beta$, it could be [10, 100, 1000]; for the positive enhancement intensity $\lambda$, it could be [50, 100, 500]; for the margin strengthening intensity $\eta$, it could be [0.1, 0.5, 1]. Further, we can expand this by automating hyperparameter tuning (like the methodology of Bayesian optimization that achieve dynamic balance between exploration and exploitation to approximate the global optimum with the fewest number of evaluations), integrating with an AutoML framework (such as the Optuna). We believe that this methodology could enable superior performance when applied to larger models and more complex datasets in engineering applications.

## H  Theoretical Perspectives on CEPA, IPA and MSPA

In this section, we present further theoretical insights into CEPA, IPA, and MSPA from the perspective of functional analysis. We begin by defining the model function space in Definition 2. Subsequently, we explicitly formulate the functionals of CEPA, IPA, and MSPA within the model function space in Definition 3.

**Definition 2** (Model Function Space). *Setting that $\mathcal{X}$ is the sample space and that $\mathcal{F}$ is the real-valued measurable function space on $\mathcal{X}$. For any function (model) $f_\theta \in \mathcal{F}$, $\forall x \in \mathcal{X}$, we have that $f_\theta(x) : \mathcal{X} \to \mathbb{R}^{w \times h}$, where $w$ and $h$ represent the width and height of the image, respectively.*

**Definition 3** (Functional Expression). *Let the preference training dataset be $\mathcal{D} = \{(x_i^w, x_i^l)\} \subset \mathcal{X} \times \mathcal{X}$. In the Hilbert space $\mathcal{F}_\mathcal{D} = L^2(\mathcal{D})$, consider the three functionals: $\mathcal{C}[f_\theta]$ (for the CEPA), $\mathcal{I}[f_\theta]$ (for the IPA) and $\mathcal{M}[f_\theta]$ (for the MSPA). Hence, for each pair sample $(x_i^w, x_i^l)$, the win ratio $X_i^w$ and the lose ratio $X_i^l$ are uniquely determined; thus, we have that:*

$$\mathcal{C}[f_\theta] = \left[\beta \cdot \log f_\theta(X_i^w) - \frac{1}{2}\right]^2 + \left[\beta \cdot \log f_\theta(X_i^l) + \frac{1}{2}\right]^2, \tag{14}$$

$$\mathcal{I}[f_\theta] = \left[\log f_\theta(X_i^w) - \log f_\theta(X_i^l) - \frac{1}{2\beta}\right]^2, \tag{15}$$

$$\mathcal{M}[f_\theta] = -\log \sigma\left[\left(\eta + 1\right)\beta \log f_\theta(X_i^w) - \beta \log f_\theta(X_i^l)\right]. \tag{16}$$

Furthermore, let us consider the Fréchet derivative of the functionals. The Fréchet derivative measures the linear variation of a functional's value in response to small perturbations of its input function. In Proposition 1, we present the calculation results of the Fréchet derivatives for CEPA, IPA and MSPA.

**Proposition 1** (Fréchet Derivative of the Functionals). *Let $f_\theta$ be a function in a Hilbert space. We introduce a random perturbation $h$ and $\epsilon$ denotes the magnitude of the perturbation. Then, under mild assumption, we can get the variation of $\mathcal{C}[f_\theta]$, $\mathcal{I}[f_\theta]$ and $\mathcal{M}[f_\theta]$ as:*

$$\delta\mathcal{C}[f_\theta](h) = \frac{\beta h(X_i^w)}{f_\theta(X_i^w)}\left(2\beta \log f_\theta(X_i^w) - 1\right) + \frac{\beta h(X_i^l)}{f_\theta(X_i^l)}\left(2\beta \log f_\theta(X_i^l) - 1\right) \tag{17}$$

$$\delta\mathcal{I}[f_\theta](h) = \left(2\log f_\theta(X_i^w) - 2\log f_\theta(X_i^l) - \frac{1}{\beta}\right)\left(\frac{h(X_i^w)}{f_\theta(X_i^w)} - \frac{h(X_i^l)}{f_\theta(X_i^l)}\right) \tag{18}$$

$$\delta\mathcal{M}[f_\theta](h) = -\left(1 - \sigma\big((\eta+1)\beta \log f(X_i^w) - \beta \log f(X_i^l)\big)\right)\left[(\eta+1)\beta\frac{h(X_i^w)}{f_\theta(X_i^w)} - \beta\frac{h(X_i^l)}{f_\theta(X_i^l)}\right] \tag{19}$$

*Proof.* **Part 1. Variation Derivation of the $\mathcal{C}[f_\theta]$.**

Firstly, introduce the random perturbation $\epsilon h$ to the $\mathcal{C}[f_\theta]$:

$$\mathcal{C}[f_\theta + \epsilon h] = \left[\beta \log \big(f_\theta(X_i^w) + \epsilon h(X_i^w)\big) - \frac{1}{2}\right]^2 + \left[\beta \log \big(f_\theta(X_i^l) + \epsilon h(X_i^l)\big) + \frac{1}{2}\right]^2.$$

For each of the two terms, we perform a Taylor expansion as follows:

$$\beta \log \big(f_\theta(X_i^w) + \epsilon h(X_i^w)\big) \approx \beta \log f_\theta(X_i^w) + \frac{\epsilon\beta h(X_i^w)}{f_\theta(X_i^w)}; \ \beta \log \big(f_\theta(X_i^l) + \epsilon h(X_i^l)\big) \approx \beta \log f_\theta(X_i^l) + \frac{\epsilon\beta h(X_i^l)}{f_\theta(X_i^l)}$$

Setting that $A = \beta \log f_\theta(X_i^w) - \frac{1}{2}$ and $B = \beta \log f_\theta(X_i^l) - \frac{1}{2}$, we have that:

$$\left[\beta \log \big(f_\theta(X_i^w) + \epsilon h(X_i^w)\big) - \frac{1}{2}\right]^2 = \left[A + \frac{\epsilon\beta h(X_i^w)}{f_\theta(X_i^w)}\right]^2 = A^2 + 2A\left(\frac{\epsilon\beta h(X_i^w)}{f_\theta(X_i^w)}\right) + \mathcal{O}(\epsilon);$$

$$\left[\beta \log \big(f_\theta(X_i^l) + \epsilon h(X_i^l)\big) + \frac{1}{2}\right]^2 = \left[B + \frac{\epsilon\beta h(X_i^l)}{f_\theta(X_i^l)}\right]^2 = B^2 + 2B\left(\frac{\epsilon\beta h(X_i^l)}{f_\theta(X_i^l)}\right) + \mathcal{O}(\epsilon).$$

Performing the differentiation to the $\epsilon$ at $\epsilon = 0$:

$$\frac{d}{d\epsilon}\mathcal{C}[f + \epsilon h]\bigg|_{\epsilon=0} = 2A \cdot \frac{\beta h(X_i^w)}{f_\theta(X_i^w)} + 2B \cdot \frac{\beta h(X_i^l)}{f_\theta(X_i^l)}.$$

Hence, the variation for $\mathcal{C}[f_\theta]$ can be expressed as:

$$\delta\mathcal{C}[f_\theta](h) = \frac{\beta h(X_i^w)}{f_\theta(X_i^w)}\left(2\beta \log f_\theta(X_i^w) - 1\right) + \frac{\beta h(X_i^l)}{f_\theta(X_i^l)}\left(2\beta \log f_\theta(X_i^l) - 1\right).$$

**Part 2. Variation Derivation of the $\mathcal{I}[f_\theta]$.**

Firstly, introduce the random perturbation $\epsilon h$ to the $\mathcal{I}[f_\theta]$:

$$\mathcal{I}[f_\theta + \epsilon h] = \left[\log \big(f_\theta(X_i^w) + \epsilon h(X_i^w)\big) - \log \big(f_\theta(X_i^l) + \epsilon h(X_i^l)\big) - \frac{1}{2\beta}\right]^2$$

Similarly, perform the Taylor expansion to $\log\big(f_\theta(X_i^w) + \epsilon h(X_i^w)\big)$ and $\log\big(f_\theta(X_i^l) + \epsilon h(X_i^l)\big)$:

$$\log\big(f_\theta(X_i^w) + \epsilon h(X_i^w)\big) \approx \log\big(f_\theta(X_i^w)\big) + \frac{\epsilon h(X_i^w)}{f_\theta(X_i^w)}; \ \log\big(f_\theta(X_i^l) + \epsilon h(X_i^l)\big) \approx \log\big(f_\theta(X_i^l)\big) + \frac{\epsilon h(X_i^l)}{f_\theta(X_i^l)}$$

Thus, we have that:

$$\log\big(f_\theta(X_i^w) + \epsilon h(X_i^w)\big) - \log\big(f_\theta(X_i^l) + \epsilon h(X_i^l)\big) = \log f_\theta(X_i^w) - \log f_\theta(X_i^l) + \epsilon\left(\frac{h(X_i^w)}{f_\theta(X_i^w)} - \frac{h(X_i^l)}{f_\theta(X_i^l)}\right)$$

Setting that $C = \log f_\theta(X_i^w) - \log f_\theta(X_i^l) - \frac{1}{2\beta}$, and we have that:

$$\mathcal{I}[f + \epsilon h] \approx \left[C + \epsilon\left(\frac{h(X_i^w)}{f_\theta(X_i^w)} - \frac{h(X_i^l)}{f_\theta(X_i^l)}\right)\right]^2 = C^2 + 2C\epsilon\left(\frac{h(X_i^w)}{f_\theta(X_i^w)} - \frac{h(X_i^l)}{f_\theta(X_i^l)}\right) + \mathcal{O}(\epsilon)$$

Performing the differentiation to the $\epsilon$ at $\epsilon = 0$:

$$\frac{d}{d\epsilon}\mathcal{I}[f + \epsilon h]\bigg|_{\epsilon=0} = 2C \cdot \left(\frac{h(X_i^w)}{f_\theta(X_i^w)} - \frac{h(X_i^l)}{f_\theta(X_i^l)}\right).$$

Hence, the variation for $\mathcal{I}[f_\theta]$ can be expressed as:

$$\delta\mathcal{I}[f](h) = \left(2\log f_\theta(X_i^w) - 2\log f_\theta(X_i^l) - \frac{1}{\beta}\right)\left(\frac{h(X_i^w)}{f_\theta(X_i^w)} - \frac{h(X_i^l)}{f_\theta(X_i^l)}\right).$$

**Part 3. Variation Derivation of the $\mathcal{M}[f_\theta]$.**

Firstly, introduce the random perturbation $\epsilon h$ to the $\mathcal{M}[f_\theta]$:

$$-\log\sigma\left[\left(\eta+1\right)\beta\log f_\theta(X_i^w + \epsilon h(X_i^w)) - \beta\log f_\theta(X_i^l + \epsilon h(X_i^l))\right].$$

Performing the Taylor expansion to $\left(\eta+1\right)\beta\log f_\theta(X_i^w + \epsilon h(X_i^w)) - \beta\log f_\theta(X_i^l + \epsilon h(X_i^l))$:

$$\approx \left(\eta+1\right)\beta\left[\log f_\theta(X_i^w) + \frac{\epsilon h(X_i^w)}{f_\theta(X_i^w)}\right] - \beta\left[\log f_\theta(X_i^l) + \frac{\epsilon h(X_i^l)}{f_\theta(X_i^l)}\right]$$

$$= \left(\eta+1\right)\beta\log f_\theta(X_i^w) - \beta\log f_\theta(X_i^l) + \epsilon\left(\left(\eta+1\right)\beta\frac{h(X_i^w)}{f_\theta(X_i^w)} - \beta\frac{h(X_i^l)}{f_\theta(X_i^l)}\right).$$

Setting that $D = \left(\eta+1\right)\beta\log f_\theta(X_i^w) - \beta\log f_\theta(X_i^l)$ and that $E = \left(\eta+1\right)\beta\frac{h(X_i^w)}{f_\theta(X_i^w)} - \beta\frac{h(X_i^l)}{f_\theta(X_i^l)}$, then:

$$\mathcal{M}[f_\theta + \epsilon h] = -\log\sigma\left(D + \epsilon E\right)$$

As we know that, $\frac{d}{du}\log\sigma(u) = \frac{1}{\sigma(u)}\sigma'(u) = 1 - \sigma(u)$; then, performing the differentiation to the $\epsilon$ at $\epsilon = 0$:

$$\frac{d}{d\epsilon}\mathcal{M}[f + \epsilon h]\bigg|_{\epsilon=0} = -\left(1 - \sigma\left(D + \epsilon E\right)\right) \cdot E\bigg|_{\epsilon=0} = -\left(1 - \sigma\left(D\right)\right) \cdot E.$$

Hence, the variation for $\mathcal{M}[f_\theta]$ can be expressed as:

$$\delta\mathcal{M}[f_\theta](h) = -\left(1 - \sigma\big((\eta+1)\beta\log f(X_i^w) - \beta\log f(X_i^l)\big)\right)\left[(\eta+1)\beta\frac{h(X_i^w)}{f_\theta(X_i^w)} - \beta\frac{h(X_i^l)}{f_\theta(X_i^l)}\right],$$

which completes the proof. $\qquad\square$

Based on the above calculation results, the following conclusion can be readily drawn.

**Fact 1.** *Intuitively, $\mathcal{M}[f_\theta]$ includes the nonlinear operator $\sigma(\cdot)$; therefore, therefore, it neither overlaps with nor is compatible with $\mathcal{C}[f_\theta]$ or $\mathcal{I}[f_\theta]$.*

Furthermore, based on the type of perturbation, we reformulate the Fréchet derivatives of $\mathcal{C}[f_\theta]$ and $\mathcal{I}[f_\theta]$ presented in Proposition 1 into vector form as follows:

$$\mathcal{C}[f_\theta] = \left(h(X_i^w), h(X_i^l)\right) \cdot \underbrace{\left(\frac{\beta}{f_\theta(X_i^w)}\left(2\beta \log f_\theta(X_i^w) - 1\right), \frac{\beta}{f_\theta(X_i^l)}\left(2\beta \log f_\theta(X_i^l) - 1\right)\right)^T}_{\vec{G}_\mathcal{C}}$$

$$\mathcal{I}[f_\theta] = \left(h(X_i^w), h(X_i^l)\right) \cdot \underbrace{\left(\frac{2\log f_\theta(X_i^w) - 2\log f_\theta(X_i^l) - 1/\beta}{f_\theta(X_i^w)}, -\frac{2\log f_\theta(X_i^w) - 2\log f_\theta(X_i^l) - 1/\beta}{f_\theta(X_i^l)}\right)^T}_{\vec{G}_\mathcal{I}}$$

Finally, based on the results above, we further elucidate the relationship between the Fréchet derivatives of $\mathcal{C}[f_\theta]$ and $\mathcal{I}[f_\theta]$, as presented in Theorem 1.

**Theorem 1.** *The Fréchet derivative of the functionals $\mathcal{C}[f_\theta]$ and $\mathcal{I}[f_\theta]$ are non-redundant (linearly independent) in the gradient space for almost every $f_\theta \in \mathcal{F}(\mathcal{D})$ that satisfies $f_\theta(X_i^w) > 0$ and $f_\theta(X_i^l) > 0$ for all $i$.*

*Proof.* As we know, if $\mathcal{C}[f_\theta]$ and $\mathcal{I}[f_\theta]$ are redundant, there must be two constants $a, b$ that makes:

$$\vec{G}_\mathcal{C} = a\vec{G}_\mathcal{I} + b\vec{1},$$

where $\vec{1}$ is the all-one-vector, representing the constant offset. Hence, there should be:

$$\frac{\beta}{f_\theta(X_i^w)}\left(2\beta \log f_\theta(X_i^w) - 1\right) = a \cdot \left(\frac{2\log f_\theta(X_i^w) - 2\log f_\theta(X_i^l) - 1/\beta}{f_\theta(X_i^w)}\right) + b \qquad (20)$$

$$\frac{\beta}{f_\theta(X_i^l)}\left(2\beta \log f_\theta(X_i^l) + 1\right) = -a \cdot \left(\frac{2\log f_\theta(X_i^w) - 2\log f_\theta(X_i^l) - 1/\beta}{f_\theta(X_i^l)}\right) + b \qquad (21)$$

For all $f_\theta(X_i^w) > 0$, $f_\theta(X_i^l) > 0$ and $f_\theta(X_i^w) \neq f_\theta(X_i^l)$, we can further derived that:

$$a = \beta \cdot \frac{f_\theta(X_i^l) \cdot (2\beta \log f_\theta(X_i^w) - 1) - f_\theta(X_i^w) \cdot (2\beta \log f_\theta(X_i^l) + 1)}{(2\log f_\theta(X_i^w) - 2\log f_\theta(X_i^l) - 1/\beta) \cdot (f_\theta(X_i^l) + f_\theta(X_i^w))}$$

$$b = \frac{2\beta^2 \left(\log f_\theta(X_i^w) + \log f_\theta(X_i^l)\right)}{f_\theta(X_i^w) + f_\theta(X_i^l)}$$

It is evident that both $a$ and $b$ are functions of $f_\theta(X_i^w)$ and $f_\theta(X_i^l)$, rather than fixed constants. Therefore, in this case, the functionals $\mathcal{C}[f_\theta]$ and $\mathcal{I}[f_\theta]$ are non-redundant.

Furthermore, for all $f_\theta(X_i^w) > 0$, $f_\theta(X_i^l) > 0$ and $f_\theta(X_i^w) = f_\theta(X_i^l) = Z$, we have that:

$$a = \beta^2$$

$$b = 2\beta^2 \cdot \frac{\log Z}{Z}$$

We can observe that $a$ is constant, whereas $b$ is not. Consequently, in this context, $\mathcal{C}[f_\theta]$ and $\mathcal{I}[f_\theta]$ are also non-redundant.

Therefore, based on the preceding discussion, we can conclude that the Fréchet derivatives of the functionals $\mathcal{C}[f_\theta]$ and $\mathcal{I}[f_\theta]$ are non-redundant (i.e., linearly independent) in the gradient space for almost every $f_\theta \in \mathcal{F}(\mathcal{D})$ that satisfies $f_\theta(X_i^w) > 0$ and $f_\theta(X_i^l) > 0$ for all $i$. This completes the proof. $\qquad \square$

# I   Results across Different Initialization Seeds

To further validate the stability and robustness of the results across different initialization seeds, we evaluate the performance of Diffusion-RainbowPA using five different random seeds (seed=40, 41, 42, 43, 44). Additionally, it is important to note in practice that the seed value used during generation and testing should be consistent with the one set during model training. For example, if the seed is set to 40 during training, the same seed value should also be set to 40 during generation and testing.

Table 11: Performance Comparison over Five Seeds on the GenEval.

| GenEval | SD-1.5 | Diffusion-DPO | SPO | Diffusion-RainbowPA | | | | | |
| | | | | Seed=40 | Seed=41 | Seed=42 | Seed=43 | Seed=44 | Average |
|---|---|---|---|---|---|---|---|---|---|
| VQAScore | 61.85 | 63.68 | 61.22 | 69.89 | 70.35 | 68.41 | 69.89 | 68.66 | **69.44** |
| CLIPScore | 33.77 | 34.52 | 33.41 | 34.04 | 35.73 | 35.01 | 35.70 | 35.42 | **35.18** |
| HPS-V2 | 26.93 | 27.17 | 27.30 | 27.52 | 27.44 | 27.45 | 27.67 | 27.64 | **27.54** |
| ImageReward | -0.200 | -0.030 | 0.083 | 0.234 | 0.322 | 0.202 | 0.359 | 0.236 | **0.271** |

Table 12: Performance Comparison over Five Seeds on the T2I-CompBench++.

| T2I-Comp++ | SD-1.5 | Diffusion-DPO | SPO | Diffusion-RainbowPA | | | | | |
| | | | | Seed=40 | Seed=41 | Seed=42 | Seed=43 | Seed=44 | Average |
|---|---|---|---|---|---|---|---|---|---|
| VQAScore | 60.71 | 62.61 | 62.73 | 68.93 | 67.73 | 67.60 | 69.20 | 67.59 | **68.21** |
| CLIPScore | 32.06 | 32.66 | 31.19 | 33.13 | 33.87 | 33.48 | 34.16 | 33.55 | **33.64** |
| HPS-V2 | 26.55 | 26.78 | 26.87 | 27.19 | 27.17 | 27.15 | 27.31 | 27.30 | **27.22** |
| ImageReward | -0.245 | -0.089 | -0.023 | 0.299 | 0.232 | 0.230 | 0.339 | 0.247 | **0.269** |

Table 13: Performance Comparison over Five Seeds on the GenAI-Bench.

| GenAI-Bench | SD-1.5 | Diffusion-DPO | SPO | Diffusion-RainbowPA | | | | | |
| | | | | Seed=40 | Seed=41 | Seed=42 | Seed=43 | Seed=44 | Average |
|---|---|---|---|---|---|---|---|---|---|
| VQAScore | 59.50 | 60.15 | 59.52 | 63.03 | 62.24 | 62.58 | 63.99 | 62.49 | **62.87** |
| CLIPScore | 33.62 | 34.11 | 32.40 | 33.75 | 34.34 | 34.47 | 34.60 | 34.33 | **34.30** |
| HPS-V2 | 26.94 | 27.15 | 27.27 | 27.56 | 27.47 | 27.53 | 27.65 | 27.62 | **27.57** |
| ImageReward | 0.237 | 0.376 | 0.360 | 0.567 | 0.490 | 0.593 | 0.584 | 0.514 | **0.550** |

Table 14: Performance Comparison over Five Seeds on the DPG-Bench.

| DPG-Bench | SD-1.5 | Diffusion-DPO | SPO | Diffusion-RainbowPA | | | | | |
| | | | | Seed=40 | Seed=41 | Seed=42 | Seed=43 | Seed=44 | Average |
|---|---|---|---|---|---|---|---|---|---|
| VQAScore | 76.28 | 77.13 | 75.61 | 79.19 | 78.88 | 79.18 | 78.66 | 77.80 | **78.74** |
| CLIPScore | 33.42 | 34.01 | 32.13 | 34.76 | 34.56 | 34.72 | 34.86 | 34.80 | **34.74** |
| HPS-V2 | 26.31 | 26.51 | 26.73 | 27.11 | 26.86 | 26.95 | 27.01 | 26.98 | **26.98** |
| ImageReward | -0.187 | -0.036 | -0.087 | 0.157 | 0.100 | 0.166 | 0.143 | 0.075 | **0.128** |

## J   Detailed Prompts for Figure 1

We summarize the detailed text prompts utilized in Figure 1 as follows (each line, from left to right):

Line 1 (on generation of animal subjects):

1. A fluffy white Samoyed dog standing in a colorful flower garden in front of a rustic house, highly detailed, digital art

2. A white puppy sitting playfully in autumn leaves, surrounded by fallen red apples, soft natural lighting

3. A cute duck wearing a chef uniform covered in cookie batter, unreal engine render 8k

4. A realistic Venusaur animal among the trees, forest lake, moss, cold weather, dark teal and amber, Sony A7 IV

5. A fluffy white cat standing on a lush green hillside under a clear sky with scattered clouds, serene natural landscape, cartoon style

6. Create an image of a cat as a gardener, wearing a straw hat, gardening gloves, and surrounded by colorful flowers

Line 2 (on generation of humanity):

1. A whimsical pink cloud-shaped building with minimalist windows and doors, floating above a vibrant blue sky with cotton-like clouds, Studio Ghibli-style animation movie texture

2. Vibrant city skyline during sunset, modern skyscrapers, colorful abstract style, warm gradient sky, digital art, urban landscape, vivid colors

3. Vibrant Christmas tree, glowing lights, abstract background, festive atmosphere, painterly style, bright and vivid color

4. Heavily decorated pile of donuts in dark red, black and gold with icing and lace trimming, dramatic lighting, dark background

5. Mystical forest with glowing mushrooms and a babbling brook

6. Album art of a hand holding a balloon emerging from the water against a red sky

Line 3 (on generation of portrait, with 3 for male and 3 for female):

1. 4d photographic image of full body image of a cute little chibi boy realistic, vivid colors octane render trending on artstation, artistic photography, photorealistic concept art, soft natural volumetric cinematic perfect light, UHD no background

2. Heroic elf warrior, golden glowing background, detailed fantasy armor, cinematic lighting, epic fantasy art, high detail

3. An intricately detailed close-up portrait of an elderly man with a long gray beard, insane face details, extremely intricate, high res, 8k, award winning

4. Extreme close-up shot portrait of a short blonde-haired beautiful woman, capturing the essence with blue eyes, lit by overhead lighting

5. A beautiful brunette pilot girl, beautiful, moody lighting, best quality, full body portrait, real picture, intricate details, depth of field, in a cold snowstorm, Fujifilm XT3, outdoors, Beautiful lighting, RAW photo, 8k, film grain, unreal engine 5, ray trace

6. Anime illustration of Princess Mononoke from Studio Ghibli, by artgerm, stunning artwork

Line 4 (on generation of futurism):

1. A dragon front face, backgrounds like hell, chain effects, angry, 3d, 8k, realistic and details, storm effective

2. A surreal alien landscape with a massive planet in the sky, rocky terrain, dramatic lighting, cinematic atmosphere, sci-fi theme, warm tones, highly detailed, 4K resolution

3. A lone astronaut standing on a volcanic landscape, detailed lava rocks, dramatic clouds in the sky, cinematic lighting, vibrant colors, realistic textures, 8k

4. Futuristic metallic humanoid robot, highly detailed face, sci-fi background, cinematic lighting, dystopian cityscape, 4K resolution

5. Futuristic cyberpunk city at night, neon lights, high-tech car, vibrant colors, cinematic lighting, highly detailed, sci-fi atmosphere, 8k resolution, unreal engine

6. A woman in black bodysuit, standing in a dark room, futurism, dramatic lighting, full-body view, cyberpunk, high contrast, detailed, 4k

