# OpenReview forum: "Diffusion-RainbowPA: Improvements Integrated Preference Alignment for Diffusion-based Text-to-Image Generation"
_TMLR — Accepted by TMLR_

### Review · Reviewer_QTKW · 2025-05-20

**Summary Of Contributions:**

This paper identifies shortcomings in current SOTA diffusion-based text-to-image alignment methods, text-image misalignment, aesthetic overfitting, and low-quality generation.

To tackle these issues, the authors introduce the Diffusion-RainbowPA, an approach integrating six enhancements addressing three key aspects: (1) calibration enhancement, (2) overfitting mitigation, and (3) performance optimization.

Experimental evaluations demonstrate that Diffusion-RainbowPA outperforms existing methods across multiple benchmarks, validating each introduced improvement through detailed ablation studies.

**Audience:**

Yes

**Broader Impact Concerns:**

The Broader Impact is well stated in page 12. It would be better to give more details of safety filtering mechanisms.

**Claims And Evidence:**

Yes

**Requested Changes:**

1. Provide explicit analysis or discussion about the computational overhead and complexity introduced by integrating multiple alignment components.

Others (But not necessary in my opinion):

2. Include results from human evaluations alongside automatic metrics to more convincingly demonstrate practical alignment effectiveness.

3. Offer guidelines or empirical insights on hyperparameter sensitivity and tuning strategies for the Diffusion-RainbowPA components.

**Strengths And Weaknesses:**

### Strengths:

1. Clearly identifies critical shortcomings (text-image misalignment, aesthetic overfitting, low-quality generation) in existing T2I alignment methods.

2. Introduces a comprehensive alignment improvement framework (Diffusion-RainbowPA) integrating calibration enhancement, overfitting mitigation, and performance optimization.

3. Demonstrates strong experimental results, significantly outperforming state-of-the-art methods across multiple benchmarks.

4. Provides thorough ablation studies, effectively validating the contribution of each individual improvement component.

5. The paper is clearly written with structured methodology, facilitating good reproducibility.

---

### Weaknesses:

1. The complexity of integrating six improvements could raise concerns about practical ease-of-use and hyperparameter tuning efforts.

2. Limited exploration of computational overhead introduced by the additional components.

---

> ### Author Response · Authors · 2025-06-08
>
> Dear Reviewer QTKW:
>
> We would like to thank you from the bottom of our heart for your time and efforts on our work. And we sincerely thank you for the thoughtful feedback that will surely turn our paper into a better shape. In the new version of our paper, additional parts have been highlighted in $\color{blue}blue$. We offer our responses to address the concerns as follows.
>
> > Q1. Provide explicit analysis or discussion about the computational overhead and complexity introduced by integrating multiple alignment components.
>
> We add the analysis and discussion in Appendix F.
>
> > Q2. Include results from human evaluations alongside automatic metrics to more convincingly demonstrate practical alignment effectiveness.
>
> We are currently in the process of applying to the Institutional Review Board (IRB) for approval of the human evaluations; and we are sorry that we are unable to present the results now. In this study, we have conducted a comprehensive evaluation of our method using ***four widely recognized and previously published*** text-image alignment metrics across ***four datasets***.
>
> > Q3. Offer guidelines or empirical insights on hyperparameter sensitivity and tuning strategies for the Diffusion-RainbowPA components.
>
> We add further discussion on hyperparameter tuning in Appendix G.
>
> > Q4. Give more details of safety filtering mechanism.
>
> We add the details in Broader Impact Statement.

---

### Review · Reviewer_uURe · 2025-05-24

**Summary Of Contributions:**

This paper introduces "Diffusion-RainbowPA," an integrated framework aimed at improving the alignment of diffusion-based text-to-image (T2I) models with human preferences. It builds upon the Diffusion-DPO framework, incorporating six distinct enhancements: (1) Step-aware Preference Alignment, (2) Calibration Enhancement Preference Alignment (CEPA), (3) Identical Preference Alignment (IPA), (4) Jensen-Shannon Divergence Constraint, (5) Margin Strengthened Preference Alignment (MSPA), and (6) SFT-like Regularization. The authors highlight the core issues of existing T2I models, specifically text-image misalignment, aesthetic overfitting, and poor image quality. Empirical evaluations confirm superior performance of Diffusion-RainbowPA across multiple standard benchmarks, such as GenEval, T2I-CompBench++, GenAI-Bench, and DPG-Bench.

**Audience:**

Yes

**Broader Impact Concerns:**

The authors acknowledge potential ethical implications and propose safety measures.

**Claims And Evidence:**

Yes

**Requested Changes:**

* Provide detailed theoretical analysis and clear discussion regarding the potential redundancy and compatibility between the CEPA, MSPA, and IPA components.

* Discuss explicitly whether the coefficients used in the combination process of Equations (12)-(13) were carefully considered, and justify why or why not these coefficients significantly impact the effectiveness of the integrated approach.

* Include additional experimental results involving human evaluators to validate the subjective quality improvements suggested by automated metrics.

* Expand the discussion on hyperparameter selection, clearly explaining the strategies and criteria for tuning each component and their interactions within the final integrated framework.

* Provide additional analysis or experiments demonstrating the stability and robustness of the results across different initialization seeds and data subsets, helping to clarify the consistency of improvements.

**Strengths And Weaknesses:**

**Strengths:**

* Clearly articulated rationale for each methodological improvement.
* Comprehensive and rigorous experimental evaluations on multiple standard datasets.
* Consistent improvements demonstrated across diverse metrics and datasets.
* Thorough ablation studies effectively validate the contribution of each proposed component.

**Weaknesses:**

* Insufficient theoretical discussion of potential overlaps or redundancies among the six methods, particularly between CEPA, MSPA, and IPA, which seem to partially address similar alignment and optimization issues.
* The manuscript lacks a theoretical analysis of the interaction and compatibility between the proposed components, potentially raising questions about their collective efficacy.
* Relies primarily on automated metrics without human evaluation to assess subjective visual quality improvements, which is essential for a method addressing aesthetic preferences.
* Experimental results do not sufficiently discuss randomness and stability across repeated trials.

---

> ### Author Response · Authors · 2025-06-08
>
> Dear Reviewer uURe:
>
> We would like to thank you from the bottom of our heart for your time and efforts on our work. And we sincerely thank you for the thoughtful feedback that will surely turn our paper into a better shape. In the new version of our paper, additional parts have been highlighted in $\color{blue}blue$. We offer our responses to address the concerns as follows.
>
> > Q1. Provide detailed theoretical analysis and clear discussion regarding the potential redundancy and compatibility between the CEPA, MSPA, and IPA components.
>
> We provide a comprehensive theoretical analysis from the perspective of functional analysis in Appendix H.
>
> > Q2. Discuss explicitly whether the coefficients used in the combination process of Equations (12)-(13) were carefully considered, and justify why or why not these coefficients significantly impact the effectiveness of the integrated approach.
>
> In this study, we improve model performance from three independent perspectives: text-image misalignment, aesthetic overfitting, and low-quality generation. In this paper, we do not include a *systematic* optimization of the coefficients, which remains an important direction for future engineering work. Within this paper, to prevent the subjective bias, we employ an equally weighted summation approach, which adheres to the classical paradigm in multi-task learning that advocates for treating tasks equally in the absence of prior knowledge [1]. On the one hand, experimental results indicate that this configuration is justified. In the Section 5.2, Diffusion-RainbowPA achieves significant improvements over SOTAs, supporting the appropriateness of the chosen coefficients. Furthermore, the ablation study reveals that removing any component leads to a decline in performance and that contributions of each component to the various metrics are complementary; therefore, the current coefficient settings preserve the synergy among the components. On the other hand, in practice, coefficients of the regularization terms are generally robust within a certain range (particularly within a given order of magnitude), and would not significantly affect the performance. However, when the magnitude of the coefficients changes, it often has a significant impact on the final performance, as further evidenced by Table 2 in work [2].
>
> > Q3. Include additional experimental results involving human evaluators to validate the subjective quality improvements suggested by automated metrics.
>
> We are currently in the process of applying to the Institutional Review Board (IRB) for approval of the human evaluations; and we are sorry that we are unable to present the results now. In this study, we have conducted a comprehensive evaluation of our method using ***four widely recognized and previously published*** text-image alignment metrics across ***four datasets***.
>
> > Q4. Expand the discussion on hyperparameter selection, clearly explaining the strategies and criteria for tuning each component and their interactions within the final integrated framework.
>
> We add further discussion on hyperparameter tuning in Appendix G.
>
> > Q5. Provide additional analysis or experiments demonstrating the stability and robustness of the results across different initialization seeds and data subsets, helping to clarify the consistency of improvements.
>
> We provide the results across five different initialization seeds in Appendix I, demonstrating the stability and robustness of our method. On the dataset selection, in order to ensure a fair comparison, we utilize the exact same dataset as SPO (CVPR 2025). In SPO, the dataset is a ***randomly selected*** subset of 4,000 prompts from the Pick-a-Pic V1 dataset. We believe that the randomly selected nature of this dataset can effectively demonstrate the consistency of improvements achieved by our method in terms of data.
>
> [1] Caruana, Rich. "Multitask learning." Machine learning 28 (1997): 41-75.
>
> [2] Sun H, Xia B, Zhao Y, et al. Positive Enhanced Preference Alignment for Text-to-Image Models[C]//ICASSP 2025-2025 IEEE International Conference on Acoustics, Speech and Signal Processing (ICASSP).

---

### Review · Reviewer_CZKt · 2025-05-26

**Summary Of Contributions:**

The paper is interesting as it derives most of the alignment and improvements from first principles. At first glance, this paper is not introducing any new model or method, but rather it is reintroducing existing solutions to problems by carefully building up from first principles. It carefully reviews most of the components, such as alignment issues, preference optimization, and loss functions supporting preference optimization.

1. Step-Aware Preference Optimization

RainbowPA adopts the SPO paradigm (Section 4.1 of the paper) as part of its framework. The authors use a step-aware preference model (sometimes a separate model that can evaluate noisy partial images) and ensure that each denoising step’s output is aligned. They also simplified the original SPO setup by using two sample trajectories instead of a larger pool (to reduce complexity and overfitting). By consistently integrating the paradigm of "step-aware preference" in RainbowPA, the model receives preference supervision at each diffusion step. By the time the diffusion reaches the final steps, the model has consistently preferred better partial solutions, so the final image is of higher quality. In contrast, a DPO-only model might accumulate errors in earlier steps (since it didn’t specifically train on them) and only correct some of them at the end.
SPO provides a much denser training signal (on every step), which can make learning more data-efficient and stable. Instead of one comparison = one gradient update, one comparison provides supervision over T steps.

2. Calibration Enhancement (CEPA)

CEPA calibration has two benefits: First, it prevents the model from becoming overconfident in its preferences. Without CEPA, the optimal solution to DPO’s logistic objective is to make the reward gap as large as possible. With CEPA, the model instead targets a moderate reward gap of 1.0 between preferred and non-preferred (0.5 – (–0.5)), which is sufficient to reflect human judgment without overshooting. Second, it improves alignment – because the model’s notion of a “win” or “loss” is kept consistent with the ±0.5 reward, the model is less likely to sacrifice prompt fidelity for the sake of marginal preference gains.

3. Overfitting Mitigation

The authors point out that models which are extensively fine-tuned tend to overfit on the trained data. To avoid this issue, the authors explore a solution from first principles of mathematical understanding.

“We circumvent the Bradley–Terry modeling assumption and employ an identical mapping to the preference function to alleviate the over-confidence.”
This means that, instead of converting reward differences into probabilities (via a sigmoid/logistic model), they train the reward difference directly to match a known value (e.g., a calibrated target).

4. Jensen-Shannon (JS) Divergence

This is an amazing idea derived from KL divergence. In simple words, KL is forward, while JS is symmetric and bounded. Using JS divergence leads to more stable and diverse optimization than KL. Since JS treats both the model and the reference more equally, the new model is not as harshly penalized for generating novel but plausible images that the original might have overlooked. This helps generative diversity.

5. Performance Optimization

DPO’s preference objective (and even with IPA) only requires that the preferred image’s log-likelihood be higher than the dispreferred’s. In practice, especially under logistic loss, a small increase is enough to classify correctly – there’s no built-in requirement for a large gap once the ordering is right. This can lead to cases where the model’s outputs for preferred vs. non-preferred samples are very close, making it brittle (a slight change could flip the preference). MSPA (Margin Strengthened Preference Alignment) introduces an explicit margin to preference alignment: the model is trained to separate the log-likelihoods (or scores) of preferred vs. dispreferred outputs by a larger amount.
Approach: MSPA modifies the preference loss to encourage a minimum gap (Δ) between the model’s treatment of winners and losers. This can be implemented in a margin-based loss style, such as hinge loss.
Enforcing a margin leads to more decisive preferences and smoother training dynamics. Gradient flow is stabilized because once the margin is satisfied, the loss gradient goes to zero (in a hinge formulation), preventing the model from endlessly increasing the gap.
In terms of outcomes, MSPA improves alignment by making the model reliably distinguish high-quality vs. low-quality generations.

6. SFT-like Regularization

RainbowPA introduces an SFT-like element for diffusion models. Pure preference training (as in DPO) only cares about relative rankings of two samples, not their absolute quality. This means if both images in a pair are poor (or both are decent), as long as one is slightly better, the model’s update is small – it doesn’t directly push the winner to be as good as it could be. SFT-like regularization addresses this by explicitly encouraging high likelihood for preferred samples in an absolute sense, beyond just beating the other sample by a tiny margin.

By injecting this supervised-like signal, the model is directly taught to increase the likelihood of human-preferred outputs instead of only worrying about their relative ranking in pairs. This yields two main benefits:

1. Higher absolute image quality – because the model is encouraged to reproduce images (or image features) that humans liked, it leans more into modes of the distribution that correspond to aesthetically pleasing or on-point images. It’s as if we fine-tuned the model on a set of high-quality generations (even though it’s implicit in comparisons).
2. Better alignment beyond the comparison baseline – the model doesn’t stop at just barely preferring the better image; it learns to make the preferred image really likely, pushing the distribution in the direction of human preference in a way similar to classical fine-tuning on a curated set. This term likely helps combat any residual tendency of the model to output mediocre images that just slightly edge out others.

**Audience:**

Yes

**Claims And Evidence:**

Yes

**Requested Changes:**

None. I really enjoyed reading the paper

**Strengths And Weaknesses:**

1. Its very intutive paper, especially reasoning of each approach is well documented.
2. They build on existing approach but provide justification of each modification step.
3. Good amount of experiments to support their hypothesis.
4. The use of Jensen-Shannon divergence and the explicit focus on promoting generative diversity over the typical KL divergence penalty is a significant contribution. It adds more stability and flexibility in optimization without compromising output quality. I am really impressed by this finding, as we all are aware of the similar improvements for stability in GAN.
5. The adoption of the SPO paradigm, with preference supervision at each diffusion step, offers a more efficient training process. This step-wise training helps mitigate errors in earlier steps, leading to better final outputs.

Weakness
1. The modifications introduced in the paper, such as enforcing a margin in MSPA and step-aware preference optimization, could face scalability issues in more complex or large-scale generative tasks. The scalability of these methods, especially when applied to higher-dimensional data or larger models, isn't fully discussed.


Note: I also appreciated the clarity of the paper and the clear evidence provided as supplementary material to verify the work.

---

> ### Author Response · Authors · 2025-06-08
>
> Dear Reviewer CZKt:
>
> We would like to thank you from the bottom of our heart for your time and efforts on our work. And we are more than delighted to receive your kind appreciation and feedback on this paper. We sincerely appreciate your positive evaluation of our work.

---

### Author Response · Authors · 2025-06-08

Dear Action Editor and Reviewers,

We would like to thank you from the bottom of our heart for your time and efforts on our work. In the new version of our paper, additional parts have been highlighted in $\color{blue}blue$. In this short note, we summarize the primary additions made to the manuscript.

* In the Appendix F, we provide further discussion on computational overhead and complexity introduced.

* In the Appendix G, we provide further discussion on hyperparameter tuning.

* In the Appendix H, we provide a comprehensive theoretical analysis from the perspective of functional analysis.

* In the Appendix I, we provide the alignment results across five different initialization seeds.

* In the section of Broader Impact Statement, we provide details on the safety filtering mechanism.

---

### Decision · Action_Editor_CVHx · 2025-07-06

**Recommendation:** Accept as is

**Audience:**

Yes

**Audience Explanation:**

Yes, this paper would definitely be of interest to a significant portion of TMLR's audience. The work addresses a highly relevant and active area of research - preference alignment for diffusion-based text-to-image models - which is at the intersection of several key ML domains including generative modeling, human preference learning, and multimodal AI.

**Claims And Evidence:**

Yes

**Claims Explanation:**

Yes, the claims are generally supported by accurate, convincing and clear evidence. The paper provides comprehensive empirical validation across four standard benchmarks (GenEval, T2I-CompBench++, GenAI-Bench, DPG-Bench) with consistent improvements demonstrated across diverse metrics, thorough ablation studies validating each component's contribution.